HYPOTHESIS

# Stochastic force generation in an isometric binary mechanical system

Vidya Murthy[1] and Josh E. Baker[1]

**Accurate models of muscle contraction are necessary for understanding muscle performance and the molecular modifications that enhance it (e.g., therapeutics, posttranslational modifications, etc.). As a thermal system containing millions of randomly fluctuating atoms that on the thermal scale of a muscle fiber generate unidirectional force and power output, muscle mechanics are constrained by the laws of thermodynamics. According to a thermodynamic muscle model, muscle's power stroke occurs with the shortening of an entropic spring consisting of an ensemble of force-generating myosin motor switches, each induced by actin binding and gated by inorganic phosphate release. This model differs fundamentally from conventional molecular power stroke models that assign springs to myosin motors in that it is physically impossible to describe an entropic spring in terms of the springs of its molecular constituents. A simple two-state thermodynamic model (a binary mechanical system) accurately accounts for muscle force–velocity relationships, force transients following rapid mechanical and chemical perturbations, and a thermodynamic work loop. Because this model transforms our understanding of muscle contraction, it must continue to be tested. Here, we show that a simple stochastic kinetic simulation of isometric muscle force predicts four phases of a force-generating loop that bifurcates between periodic and stochastic beating through mechanisms framed by two thermodynamic equations. We compare these model predictions with experimental data including observations of spontaneous oscillatory contractions (SPOCs) in muscles and periodic force generation in small myosin ensembles.**

## Introduction

Muscle is a complex and dynamic macromolecular system that is integral to physiological functions such as locomotion, digestion, and the beating heart. While high-resolution structures and single-molecule mechanics have provided significant insights into molecular structure–function relationships (Guilford et al., 1997; Finer et al., 1994; Molloy et al., 1995; Baker et al., 2002; Lymn and Taylor, 1971; Cooke, 1997; Goldman, 1987), the mechanisms of muscle's structure–function relationships such as muscle's power stroke (the work performed by shortening muscle) remains unclear (Baker and Thomas, 2000a, 2000b; Baker, 2022a, 2022b, 2023a). In 1998, we observed within isometric muscle an ensemble of myosin force-generating switches or discrete lever arm rotations (Baker et al., 1998), each induced by actin binding and gated by the release of inorganic phosphate (Fig. 1 A). In 1999, we showed that this ensemble of switches responds to changes in muscle force like an entropic spring (Baker et al., 1999), and in 2000, we proposed that the shortening of this entropic spring is the muscle's power stroke mechanism (Baker and Thomas, 2000a). This differs fundamentally from the conventional description of a molecular power stroke mechanism (Huxley, 1957) that occurs with the shortening of a molecular spring.

We have since developed a simple two-state thermodynamic model of muscle force generation that accurately accounts for the observed muscle force–velocity relationship (Baker and Thomas, 2000a; Baker, 2022b) and the four phases of muscle force transients following a rapid perturbation to either muscle chemistry or muscle force (Baker, 2022b). This simple model also accounts for muscle's thermodynamic work loop and has broad implications for physical chemistry, suggesting a novel thermodynamic kinetic formalism (Baker, 2023b, Preprint), a solution to the Gibbs (mixing) paradox (Baker, 2023c), quantized thermodynamics (Baker, 2024a, Preprint), and constructive entropic forces in biological systems (Baker, 2024a, Preprint). Here, we describe novel model predictions for stochastic isometric force generation, showing that in stochastic simulations, four phases of a thermodynamic force-generating loop emerge bifurcated between periodic and stochastic beating.

Most models of muscle contraction to date are corpuscular mechanic models that assume that muscle force is determined from the forces of individual myosin motors (Baker, 2023a). In contrast, in a thermodynamic muscle model, muscle force is determined from the free energy of a myosin motor ensemble

[1]Department of Pharmacology, Reno School of Medicine, University of Nevada, Reno, NV, USA.

Correspondence to Josh E. Baker: jebaker@unr.edu.

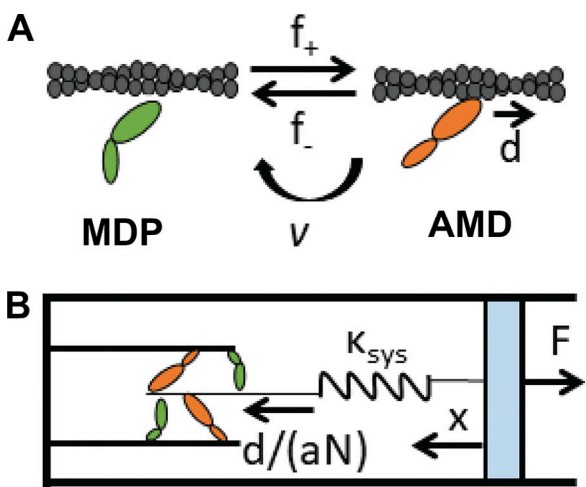

Figure 1. **A binary mechanical model. (A)** Two-state scheme in which actin (gray helix) binding to a single myosin motor (ovals) in the MDP state (green) with bound ADP, D, and phosphate, P, induces a myosin lever arm rotation upon P release that displaces elements external to that myosin a distance, *d*, to form the AMD state (orange). The reversible binding reaction occurs with forward, $f_+$, and reverse, $f_-$, rates. Myosin motors irreversibly detach from actin through an active (ATP-dependent) process at a rate *v*. **(B)** Muscle force is represented by a single spring with stiffness $\kappa_{sys}$ that on one end (left) is reversibly stretched a distance $d/(aN)$ with each binding step while the other end is defined by macroscopic mechanics (here held at a fixed force or length).

(Baker, 2022b; Baker and Thomas, 2000a). These two theories are mutually exclusive because the entropic contribution to free energy does not exist within individual myosin motors. Because a thermodynamic model transforms our understanding of muscle contraction, it is imperative to establish and test thermodynamic mechanisms of muscle contraction. Here, we continue this 25-year effort (Baker, 2023a) by establishing the thermodynamic mechanisms of isometric muscle force generation.

A simple kinetic simulation of the binding reaction in Fig. 1 A accurately describes most key aspects of muscle contraction (Baker and Thomas, 2000a; Baker, 2022b, 2023a, 2024b). Force generation by a non-equilibrium binding reaction is described by a simple spring equation (Eq. 1), where a system spring is stretched by a binding reaction with rates defined by the force-dependent binding free energy equation (Eq. 2). Eq. 1 provides a mathematical solution from which Hill's force–velocity relationship is derived (Baker and Thomas, 2000a; Baker, 2022b). The time course of force generation can be determined either by solving differential equations (continuous) or through stochastic time steps (discrete). We previously used Matlab to generate continuous time courses of muscle force and state occupancies (Baker and Thomas, 2000a; Baker, 2022b, 2024b). However, these models do not capture the emergent stochastic mechanics of myosin motor ensembles. Here, we use Python to generate discrete changes in muscle force and state occupancies (Data S1). We observe that, framed by Eqs. 1 and 2, isometric force generation follows four phases (binding, ergodic, isothermal, and catastrophic) that create a force-generating loop as observed in small myosin motor ensembles (Hwang et al., 2021; Pertici et al.,

2018) and in spontaneous oscillatory contractions (SPOCs) of the muscle (Fabiato and Fabiato, 1978). A parametric analysis shows how the periods and amplitudes of these loops depend on a limited number of parameters and how under certain conditions these loops bifurcate between stochastic and periodic beating. This model reconciles disparate force-generating behaviors observed in different in vitro force studies, makes clear predictions about the effects of ligand concentrations, system stiffness, binding kinetics, etc. on force generation, and has novel implications for mechanistic differences between tonic and phasic muscle.

## Materials and methods

Here, we simulate isometric force generation using the thermodynamic model of force generation illustrated in Fig. 1 and formally developed in Baker (2022b, 2023b, *Preprint*). Single-molecule mechanic studies show that the formation of a strong bond between actin (A) and myosin (M) induces a discrete conformational change (a myosin lever arm rotation) in individual myosin motors that displaces elastic elements external to the motor a distance, *d*, of 8 nm (Baker et al., 1998; Baker and Thomas, 2000a; Baker, 2022a; Sweeney et al., 2020). This force-generating molecular switch occurs through an intermediate step in the actin–myosin catalyzed ATP hydrolysis reaction, where actin binding to a myosin motor in the MDP state (myosin, M, with bound ADP, D, and inorganic phosphate, P) is gated by the release of P in forming the AMD state (Baker et al., 1998, 2002). This binding reaction is reversible with forward, $f_+$, and reverse rates, $f_-$, and is pulled from equilibrium through the ATPase reaction which irreversibly transfers myosin motors from AMD to MDP at the effective rate, *v*, for ADP release, ATP-induced actin–myosin dissociation, and ATP hydrolysis. Here, we assume ATP hydrolysis is not rate-limiting.

Our computational model is nothing more than a stochastic kinetic simulation of a force-generating binding reaction (a spring equation, Eq. 1) with forward and reverse binding rates defined in terms of the binding free energy (Eq. 2). The collective force generated when the binding reaction stretches a system spring of stiffness, $\kappa_{sys}$, is

$$F = \kappa_{sys} \cdot d \cdot N_{AMD} / (aN),$$

where $d/(aN)$ is the spring displacement per binding step, $N_{AMD}$ is the number of bound motors (i.e., the net number of steps), and N is the total number of myosin motors (Baker, 2022b). For many reasons (see below), the binding reaction can equilibrate with a force, F, that differs from the equilibrium force, $F_o$, and ergodicity, a, is the ratio of these forces, $a = F/F_o$. As described in Baker (2022b, 2024b), the force generated by a step, d, of an individual motor equilibrates with and thus is distributed among all N myosin motors. This is analogous to a displacement, d, by one of N equilibrated parallel springs that displaces the effective spring a distance d/N. When on average, a fraction, a, of myosin motors equilibrate with the equilibrium force, d is distributed among aN motors. When all motors equilibrate, the displacement is d/N. When only one of N motors equilibrates, the displacement is d (this is the assumption in corpuscular

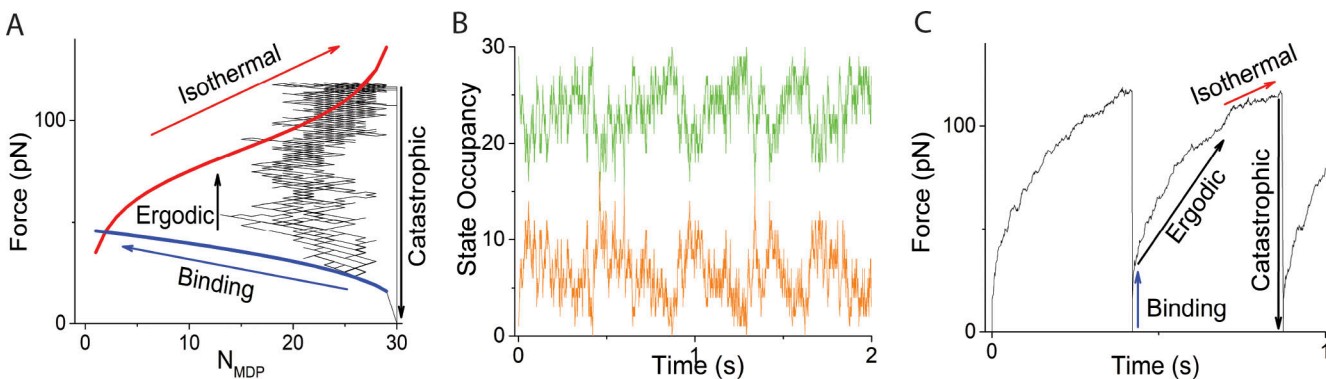

**Figure 2. Simulations using $N$ = 30, $\kappa_{sys}$ = 2 pN nm$^{-1}$, and $v$ = 50 s$^{-1}$ compared with Eqs. 1 and 2. (A–C)** Plots of Eqs. 1 (blue curve) and 2 (red curve) are overlaid with replots (black trace) of simulated time courses of (B) $N_{MDP}$ (green trace), $N_{AMD}$ (orange trace), and (C) force. Four phases of isometric force generation (binding, ergodic, isothermal, and catastrophic) are labeled both in the $N_{MDP}$ domain of force (Eqs. 1 and 2, A) and in the time domain of force (C). In these simulations, $\Delta G^o$ = −5.7 RT and $d$ = 8 nm which according to Eq. 2 gives $F_o$ = −$N\Delta G^o/d$ = 90 pN when $N_{AMD}$ = $N_{MDP}$.

mechanic models where force generation is not distributed among motors) (Huxley, 1957). Thus, values for $a$ range from $1/N$ to 1.

Unlike in Baker (2022b), here, we substitute $a = F/F_o$ into the above equation to obtain

$$F = \sqrt{\frac{\kappa_{sys}dN_{AMD}F_o}{N}}, \qquad (1)$$

From this equation, the time course of $F$ can be determined from a simple kinetic simulation of the binding reaction (the time course of $N_{AMD}$). Eq. 1 is plotted in Fig. 2 A (blue line).

When the binding reaction equilibrates, Eq. 1 no longer defines $F$. Instead, $F$ is defined by the binding free energy equation derived in Baker (2022b) as

$$\Delta_r G = \Delta G^o + k_B T \cdot \ln(N_{AMD}/N_{MDP}) + F \cdot d/(aN),$$

where $\Delta_r G$ is the binding reaction free energy, $\Delta G^o$ is the standard binding free energy, the $k_B T \cdot ln$ term is a change in entropic energy of the spring (Baker, 2022b, 2023b, *Preprint*), and $N_{MDP}$ and $N_{AMD}$ are the numbers of motors in the bound and unbound states. Here, for large $N_{AMD}$, the $N_{AMD} + 1$ term in Baker (2022b, 2023b, *Preprint*) is approximated as $N_{AMD}$. If the binding reaction equilibrates in an ergodic state ($a = 1$), an equilibrium ($\Delta_r G = 0$) force, $F_o$, is reached

$$F_o = -N[\Delta G^o + k_B T \cdot \ln(N_{AMD}/N_{MDP})]/d. \qquad (2)$$

This equation is plotted in Fig. 2 A (red line).

When the binding reaction equilibrates with a non-equilibrium force ($F < F_o$) in a non-ergodic state ($a < 1$), substituting Eq. 2 into the free energy equation gives $\Delta_r G = (F_o d/N - Fd/aN)$. We have shown that this equation forms the basis for Hill's muscle force–velocity relationship (Baker and Thomas, 2000a), where the binding reaction equilibrates with a non-equilibrium isotonic force, $F = aF_o$ and the free energy available for work is relative to the equilibrium force ($a = 1$). We have shown that during unloaded muscle shortening, intermolecular interactions generate a non-ergodic frictional force, $F_f$, against which muscle shortens, in which case $F_f = aF_o$, where $aF_o$ is Hill's coefficient of shortening heat (Baker, 2022b). We have shown that during

phase 2 (the binding reaction) of a transient force response to a chemical or mechanical perturbation of isometric force, the binding reaction can generate force in series elastic elements that result in a non-equilibrium force, $F = aF_o$, where the phase 3 response occurs when $F$ approaches $F_o$. Here, we show that the force, $F$, generated in isometric muscle upon equilibration of the binding reaction (Fig. 2 A, binding phase) can be less than $F_o$ ($a < 1$).

As described in Baker (2023b, *Preprint*) (which uses a nomenclature slightly different from that in Baker [2022b]), the probability of a forward binding step relative to a reverse binding step, $f_+/f_-$, is determined by the binding free energy, where $f_+$ and $f_-$ are the forward and reverse binding rates, respectively. Specifically,

$$\frac{f_+}{f_-} = e^{\frac{\frac{-Fd}{aN} - k_B T \ln \frac{N_{AMD}}{N_{MDP}} - \Delta G^o}{k_B T}}, \qquad (3)$$

where the binding free energy is in the exponent. According to Eq. 3, when the binding reaction equilibrates ($f_+ = f_-$), net force generation through Eq. 1 stops because there is no net change in $N_{AMD}$. In our simulations, the ATPase reaction continuously pulls the binding reaction out of equilibrium, which upon re-equilibration actively generates additional force.

In short, our computer modeling is nothing more than a simple kinetic simulation of a binding reaction (Eq. 1) with forward and reverse binding rates (from Eq. 3)

$$f_+ = f_+^o N_{MDP} e^{-0.5\left(\frac{\frac{Fd}{aN}}{k_B T}\right)}$$

and

$$f_- = f_-^o N_{AMD} e^{0.5\left(\frac{Fd/aN}{k_B T}\right)}$$

where $f_+^o$ and $f_-^o$ are forward and reverse rate constants, respectively, and the partitioning between forward and reverse rates is arbitrarily chosen to equal 0.5.

Four phases of isometric force generation emerge from this simple kinetic simulation (Fig. 2 A). The first phase (binding

phase, Eq. 1) is the binding reaction which occurs at the binding relaxation rate and terminates when the binding reaction equilibrates. Both the second and third phases occur when the ATPase reaction perturbs the binding reaction from equilibrium which upon re-equilibration generates additional force at a rate $v$. In a non-ergodic state ($a < 1$) this active force generation increases ergodicity, $a = F/F_o$ (ergodic phase). In an ergodic state ($a = 1$), active force generation increases $F_o$ along the binding isotherm (isothermal phase) corresponding to a decrease in $N_{AMD}$ (Eq. 2). The isothermal phase is not somehow prescribed in our simulations; it emerges from the definition of binding rate constants in terms of binding free energy, which forces the reaction to equilibrate along the binding isotherm. When $N_{AMD}$ equals zero, force can no longer be maintained, and $F$ is set to zero (catastrophic phase), completing the loop.

Our stochastic simulations of the binding reaction (Fig. 1) are developed using Python. Briefly, the state of each of $N$ myosin motors is stored in an array. To determine if a motor leaves its current state, a random number is generated for each motor with each microsecond time step and compared with the net rate out of a motor's current state. If a motor leaves a state, a second random number is generated and compared with the relative rates of multiple pathways out of a motor's current state to determine which of those pathways the motor takes. In all simulations, we used model parameters consistent with experimental studies: $f_+^o = 30\ s^{-1}$ and $f_-^o = 0.1\ s^{-1}$ (rate constants that give a $\Delta G^o$ of $-5.7$ RT) and $d = 8$ nm (Woledge et al., 1985; Guilford et al., 1997).

### Online supplemental material

Fig. S1 shows the supplement to simulations in Fig. 3 varying system spring stiffness, $\kappa_{sys}$, using $v = 0$, $N = 30$. Fig. S2 shows the supplement to simulations in Fig. 4 varying $N$, using $v = 0$ and $\kappa_{sys} = 2$ pN/nm. Fig. S3 shows the supplement to simulations in Fig. 5, varying $v$, using $N = 30$ and $\kappa_{sys} = 2$ pN/nm. Fig. S4 shows the supplement to simulations in Fig. 6 varying $\kappa_{sys}$, using $v = 50\ s^{-1}$ and $N = 30$. Fig. S5 the shows supplement to simulations in Fig. 7 varying $N$, using $v = 50\ s^{-1}$ and $\kappa_{sys} = 2$ pN/nm. Data S1 provides the sample python code used for simulations.

## Results

Here, we performed simple stochastic kinetic simulations of a force-generating binding reaction (Eq. 1) with binding rates defined in terms of the binding free energy (Eq. 3) to obtain time courses of the occupancy of the intermediate states AMD ($N_{AMD}$, Fig. 2 B, orange trace) and MDP ($N_{MDP}$, Fig. 2 B, green trace) in the actin–myosin ATPase reaction and the corresponding time courses of $F$ (Fig. 2 C). These time courses (Fig. 2, B and C) are then replotted as $F$ versus $N_{MDP}$ (Fig. 2 A) so that they can be compared with Eqs. 1 (blue line) and 2 (red line).

These simulations result in up to four phases of isometric force generation (Fig. 2 A). Starting at $F = 0$ and $N_{AMD} = 0$, the non-equilibrium binding reaction (binding phase) generates force at a rate $f_+ + f_-$. If this is much faster than the ATPase rate, $v$, the binding reaction equilibrates before the ATPase reaction begins to pull the binding reaction from equilibrium at a rate, $v$,

generating an active force upon re-equilibration of the binding reaction. If the binding reaction equilibrates at a non-equilibrium ($F < F_o$), non-ergodic ($a < 1$) force, then active force generation increases $a = F/F_o$ (ergodic phase) until $a = 1$. At an equilibrium ($F = F_o$) force, active force generation increases $F_o$ along the binding isotherm (isothermal phase), corresponding to a decrease in $N_{AMD}$ (Eq. 2). This occurs, not because it is explicitly prescribed in our simulations, but because binding rates that are defined in terms of the binding free energy (Fig. 3 and Fig. S1) constrain equilibration along the isotherm. In other words, a decrease in $N_{AMD}$ is required to balance the equilibrium free energy equation when $F_o$ increases through active force generation. The isothermal phase terminates when $N_{AMD} = 0$, at which point there are no bound myosin motors to maintain force, and $F$ is set to zero (catastrophic phase). This returns the simulation back to the starting parameters $N_{AMD} = 0$ and $F = 0$, completing a force-generating loop that then repeats.

The simulated force-generating loop is framed by Eqs. 1 and 2 (Fig. 2 A); however, because these equations describe ideal thermodynamic processes, under non-ideal conditions, the simulated force-generating loop need not follow these equations. For example, because binding rates are defined in terms of the binding free energy, force generation during the binding phase (Eq. 1) is influenced by Eq. 2. As another example, if the ATPase rate, $v$, is comparable to the binding relaxation rate, the binding reaction never reaches a non-ergodic equilibrium, and the ergodic phase is held far from equilibrium (see below). Here, we perform a parametric analysis of this force-generating loop. We begin by simulating isometric force generation through equilibration of the binding reaction alone ($v = 0$).

The system spring stiffness, $\kappa_{sys}$, is highly variable and is determined by many factors including protein stiffness, the number of actin-bound myosin motors, structural proteins in the contractile assembly, as well as the extracellular matrix (Westerhof et al., 2006; Huxley and Tideswell, 1996). In Fig. 3, we consider the effects of the system spring stiffness, $\kappa_{sys}$, on the binding reaction. The left panels in Fig. 3 are simulated time courses of both force, $F$, and ergodicity, $a$, (inset), and the right panels are $F$ versus $N_{MDP}$ from the same simulations overlaid with Eqs. 1 and 2. Panels from top to bottom are simulations performed for $\kappa_{sys}$ values ranging from 0.01 to 10 pN/nm.

When the system spring stiffness is low (Fig. 3, A and B, $\kappa_{sys} = 0.01$ pN/nm), myosin motors rapidly bind actin with no significant force generation such that $a$ never exceeds $1/N$. Because the total force, $F$, generated by $N$ myosin motors never exceeds the equilibrium force of one myosin motor, $F_o/N$, the maximum system force is unable to balance the chemical force of even one motor. Thus, the reaction proceeds to $N_{AMD} = N$ as it would in solution in the absence of mechanical force.

At higher stiffnesses (Fig. 3, C and D, $\kappa_{sys} = 0.16$ pN/nm), the value of $a$ rapidly exceeds $1/N$, and the binding reaction and corresponding force generation equilibrate at a non-ergodic isotherm when $N_{AMD} = N_{MDP}$. This results from Eq. 3, which shows that starting from $N_{AMD}/N_{MDP} = 1$, with a reverse binding step the decrease in both $F$ and $N_{AMD}$ favors net force generation $f_+ > f_-$ and with a forward binding step the increase in $F$ and $N_{AMD}$ favors a net reversal of force generation, $f_+ < f_-$. This

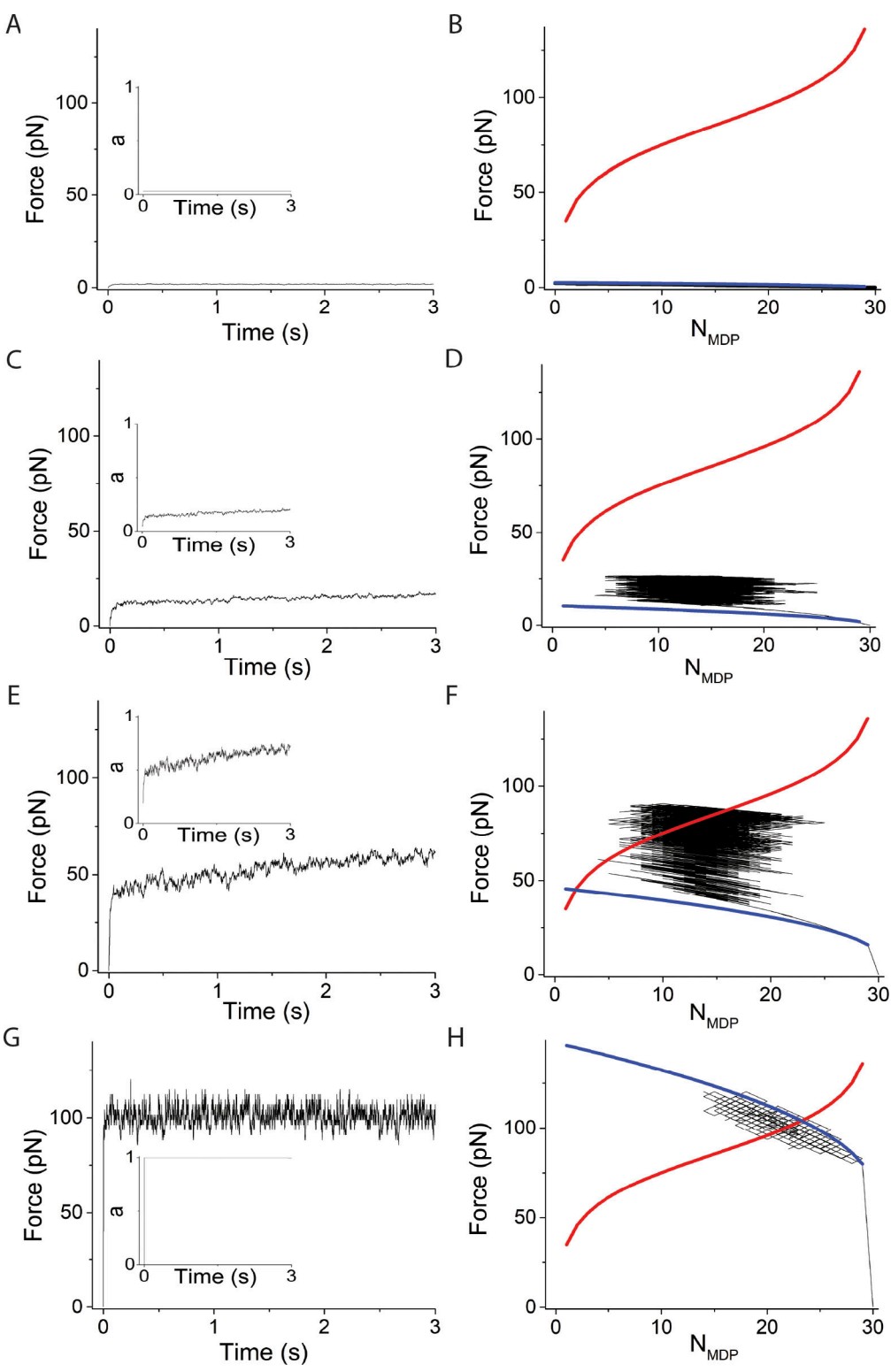

Figure 3. **Simulations varying system spring stiffness, $\kappa_{sys}$, using $v$ = 0, $N$ = 30. (A–H)** Left panels are simulated time courses of force, $F$, and ergodicity, $a$ (inset), and right panels are $F$ and $N_{MDP}$ replotted and overlaid with Eqs. 1 (blue curve) and 2 (red curve) at $\kappa_{sys}$ values of (A and B) 0.01 pN/nm, (C and D) 0.16 pN/nm, (E and F) 2 pN/nm, and (G and H) 10 pN/nm. In these simulations, $\Delta G^o$ = −5.7 RT and $d$ = 8 nm, which according to Eq. 2 gives $F_o$ = −$N\Delta G^o/d$ = 90 pN when $N_{AMD}$ = $N_{MDP}$. The simulated time courses of state occupancies are in Fig. S1.

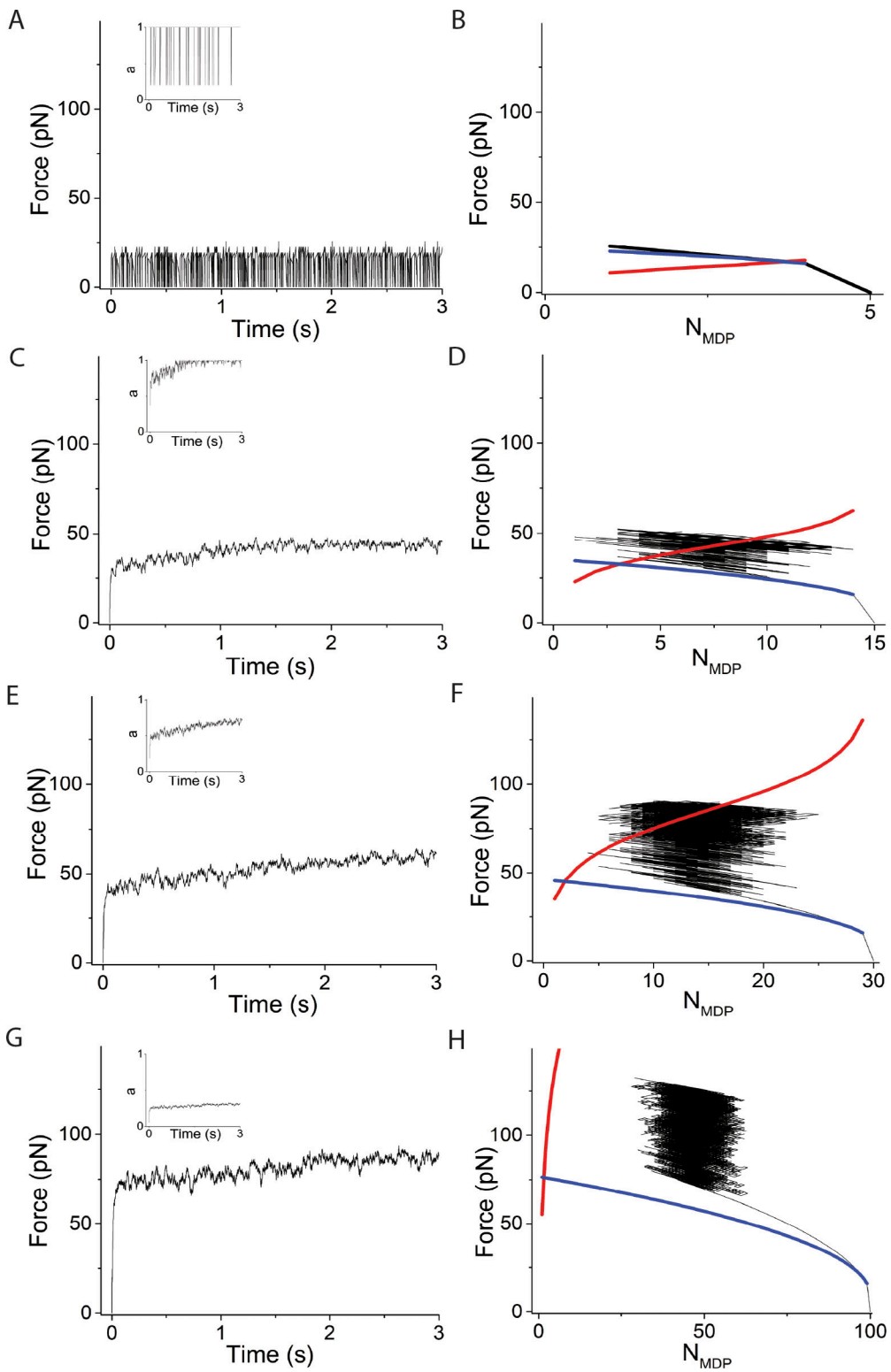

Figure 4.   **Simulations varying *N*, using *v* = 0 and κ$_{sys}$ = 2 pN/nm. (A–H)** Left panels are simulated time courses of force, *F*, and ergodicity, *a* (inset), and right panels are *F* and *N$_{MDP}$* replotted and overlaid with Eqs. 1 (blue curve) and 2 (red curve) using *N* values of (A and B) 5, (C and D) 15, (E and F) 30, and (G and H) 100. In these simulations ΔG° = −5.7 RT and *d* = 8 nm which according to Eq. 2 gives *F$_o$* = −*N*ΔG°/*d* = 3N pN when *N$_{AMD}$* = *N$_{MDP}$*. The simulated time courses of state occupancies are in Fig. S2.

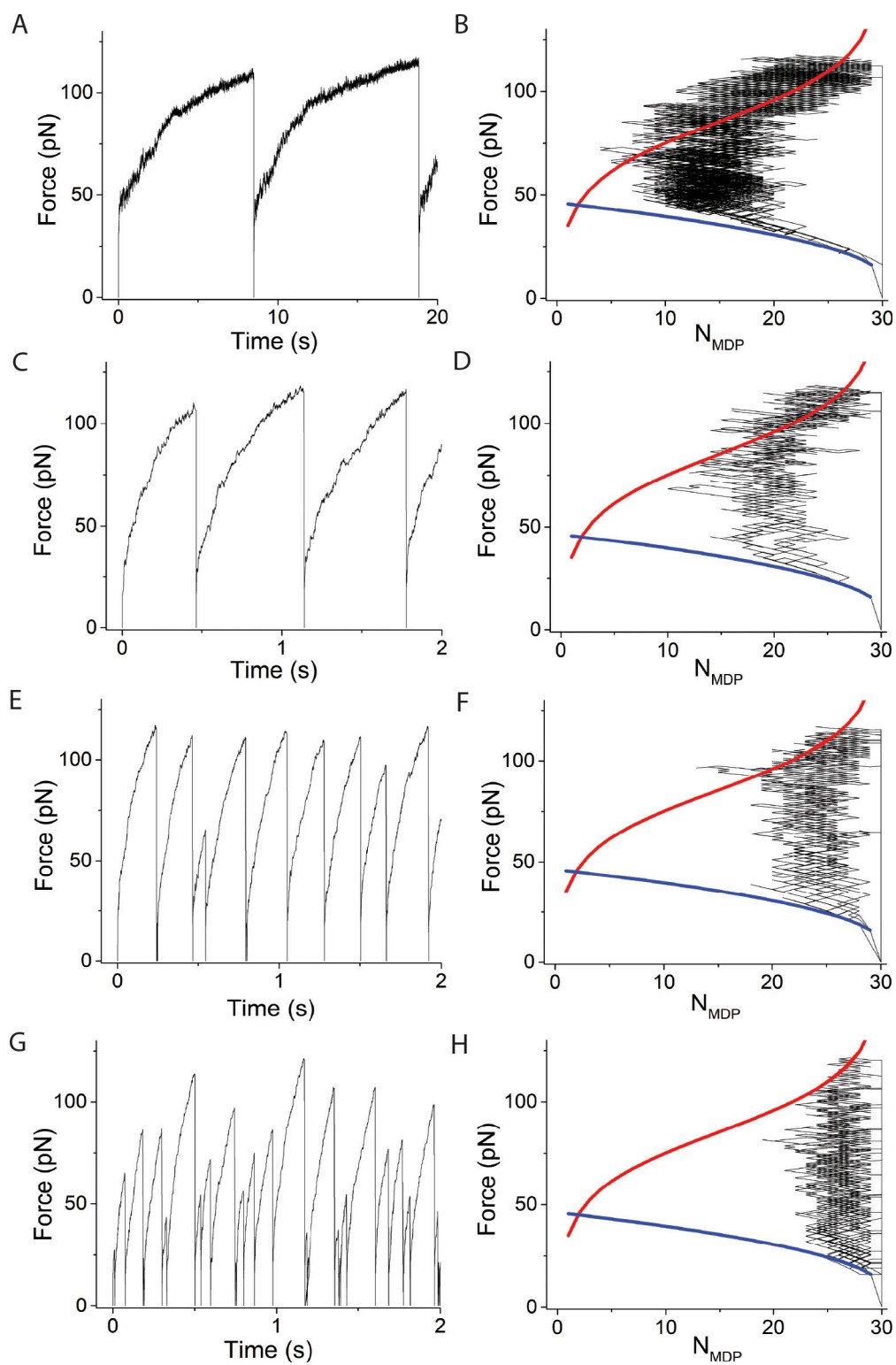

Figure 5.   **Simulations varying *v*, using *N* = 30 and κ$_{sys}$ = 2 pN/nm. (A–H)** Left panels are simulated time courses of force, *F*, and right panels are *F* and *N*$_{MDP}$ replotted and overlaid with Eqs. 1 (blue curve) and 2 (red curve) using *v* equal to (A and B) 1 s$^{-1}$, (C and D) 25 s$^{-1}$, (E and F) 100 s$^{-1}$, and (G and H) 200 s$^{-1}$. In these simulations, ΔG° = −5.7 RT and *d* = 8 nm which according to Eq. 2 gives *F*$_o$ = −*N*ΔG°/*d* = 90 pN when *N*$_{AMD}$ = *N*$_{MDP}$. The simulated time courses of state occupancies are in Fig. S3.

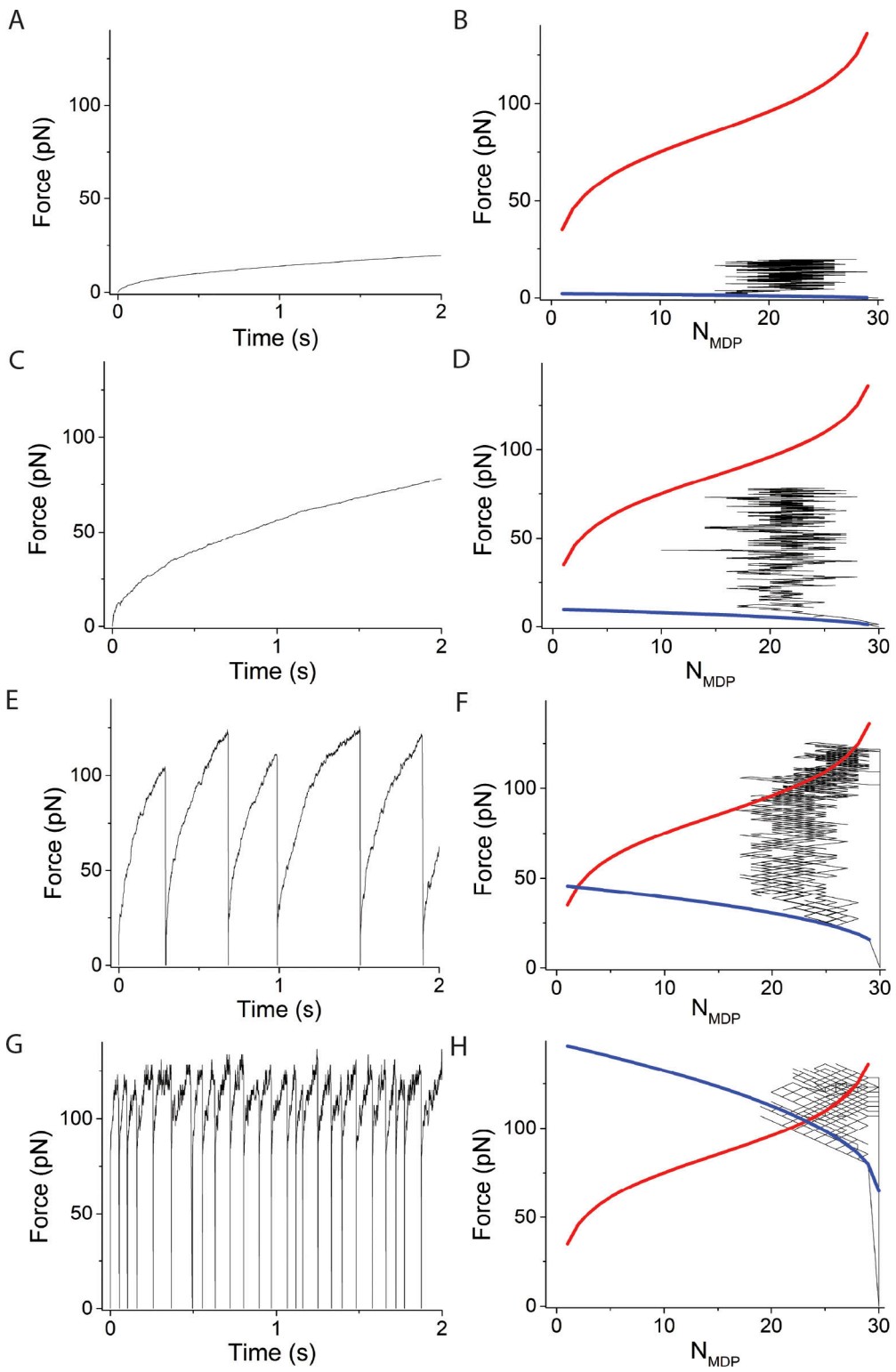

Figure 6.   **Simulations varying κ$_{sys}$, using *v* = 50 s$^{-1}$ and *N* = 30. (A–H)** Left panels are simulated time courses of force, *F*, and right panels are *F* and N$_{MDP}$ replotted and overlaid with Eqs. 1 (blue curve) and 2 (red curve) using κ$_{sys}$ values of (A and B) 0.01 pN/nm, (C and D) 0.16 pN/nm, (E and F) 2 pN/nm, and (G and H) 10 pN/nm. In these simulations, ΔG° = −5.7 RT and *d* = 8 nm which according to Eq. 2 gives *F$_o$* = −NΔG°/*d* = 90 pN when N$_{AMD}$ = N$_{MDP}$. The simulated time courses of state occupancies are in Fig. S4.

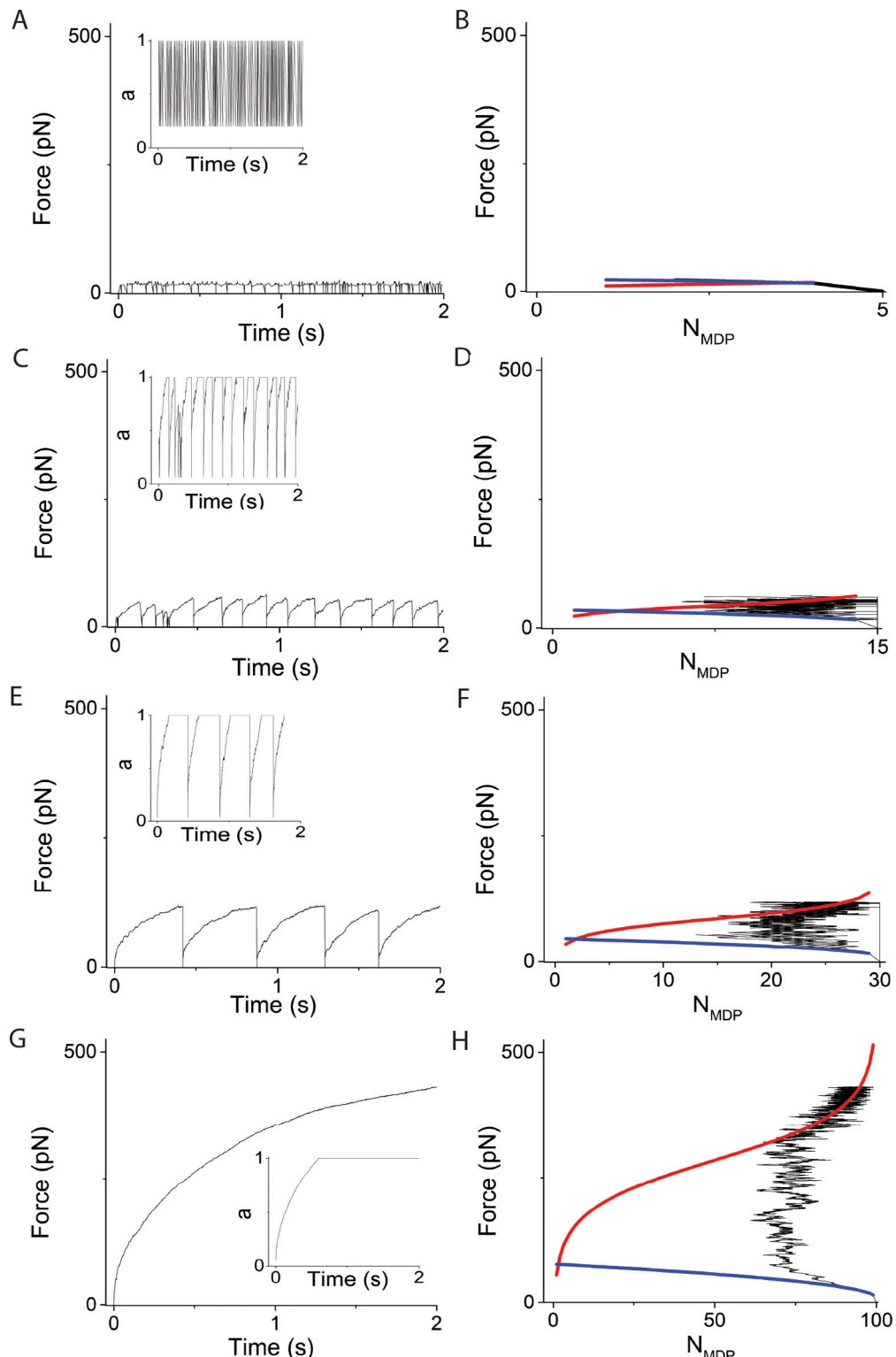

Figure 7. **Simulations varying *N*, using *v* = 50 s⁻¹ and κ_sys = 2 pN/nm. (A–H)** Left panels are simulated time courses of force, *F*, ergodicity, *a* (inset), and right panels are *F* and $N_{MDP}$ replotted and overlaid with Eqs. 1 (blue curve) and 2 (red curve) using *N* values of (A and B) 5, (C and D) 15, (E and F) 30, and (G and H) 100. In these simulations, ΔG° = −5.7 RT and *d* = 8 nm which according to Eq. 2 gives $F_o = −NΔG°/d$ = 3N pN when $N_{AMD} = N_{MDP}$. The simulated time courses of state occupancies are in Fig. S5.

effectively creates a restoring force that maintains maximum entropy, $N_{AMD} = N_{MDP}$ (Baker, 2023b, *Preprint*). This is the case independent of values for $f_+$ and $f_-$ in the absence of force, as experimentally observed (Baker et al., 1999). In our simulations, spontaneous ergodic force generation is observed where $F$ approaches $F_o$. This is because the effective displacement, $d/(aN)$, by a forward step is larger than that of the reversal of that step (the latter occurs at a higher force and larger ergodicity, $a$). At sufficiently high stiffness, adiabatic force generation equilibrates with the equilibrium isotherm in an ergodic state (Fig. 3, G and H).

In Fig. 4 and Fig. S2, we consider the effects of the number, $N$, of myosin motors on simulated binding reactions. The left panels are plots of the simulated time course for both force, $F$, and ergodicity, $a$, (inset), and the right panels are $F$ versus $N_{MDP}$ from the same simulations overlaid with Eqs. 1 and 2. Panels from top to bottom are simulations performed with values for $N$ ranging from 5 to 100 myosin motors. At low $N$ ($N = 5$), adiabatic force generation directly equilibrates with the equilibrium isotherm with no ergodic phase. The ergodic force required to reach the equilibrium isotherm increases with $N$.

Fig. 5 and Fig. S3 show simulations of the model in Fig. 1 introducing an ATPase reaction that cycles at a rate, $v$. The left panels are plots of the simulated time course of both force, $F$, and ergodicity, $a$, (inset), and the right panels are $F$ versus $N_{MDP}$ from the same simulation overlaid with Eqs. 1 and 2. Panels from top to bottom are simulations with values for $v$ increasing from 1 to 200 sec$^{-1}$.

Force generation simulated at low $v$ not surprisingly resembles simulations in Fig. 4 with $v = 0$, where the binding reaction and corresponding force generation equilibrate at a non-ergodic force when $N_{AMD} = N_{MDP}$. With increasing $v$, the ATPase reaction pulls the binding reaction further from a non-ergodic equilibrium ($N_{AMD} < N_{MDP}$) in a non-equilibrium steady state. During this steady state, for every myosin motor irreversibly transferred from AMD to MDP through the ATPase reaction, one myosin motor generates force through the binding reaction, increasing the ergodicity, $a = F/F_o$ until $a = 1$. At this point, $a$ is set to 1, and force generation increases $F_o$ along the binding isotherm (Eq. 2), corresponding to a decrease in $N_{AMD}$. Isothermal force generation continues until the last bound myosin motor detaches, at which point a catastrophic loss of force, $F$, returns the system to its initial state. This four-phase force-generating loop repeats, resulting in periodic force generation. The maximum amplitude of periodic force generation is the maximum force along the isotherm (Eq. 2, $N_{MDP} = N - 1$) with a period that varies with the inverse ATPase rate, $v$. The maximum periodic amplitude is unaffected by $v$ because the force along the isotherm at which the last bound myosin motor detaches is independent of $v$ (right panels).

However, at sufficiently high rates, $v$, the force-generating loop is right-shifted to a point where ergodic force generation occurs when most motors are detached (small $N_{AMD}$), increasing the probability that the few remaining bound motors spontaneously detach before $F$ reaches the isothermal force, $F_o$. This results in stochastic force-generating loops with stochastic amplitudes and durations that are smaller than those of periodic

loops. In Fig. 5, E and F ($v = 100$ s$^{-1}$), one out of nine force-generating loops terminates through this stochastic mechanism with a smaller period and amplitude. In Fig. 5, G and H ($v = 200$ s$^{-1}$), most force-generating loops terminate stochastically and exhibit stochastic periods and amplitudes. The probability of stochastic periodic force generation increases at low $N$, low $f_+$, high $\kappa_{sys}$, and high $f_-$.

Fig. 6 and Fig. S4 show that increasing $\kappa_{sys}$ decreases the period but not the amplitude of force-generating loops because increasing $\kappa_{sys}$ increases the steepness of the force-generating binding reaction (Eq. 1 and Fig. 6 right panels blue curves) without affecting the isotherm (Eq. 2 and Fig. 6 right panels red curves), creating smaller loops (shorter periods) of the same amplitude. Fig. 7 and Fig. S5 show that increasing $N$ decreases the period and increases the amplitude of force-generating loops because increasing $N$ increases the isotherm (Eq. 2 and Fig. 7 right panels red curves), creating larger loops (longer periods) with larger amplitudes.

## Discussion

In almost every model of muscle contraction to date, muscle force is determined from myosin motor forces, where the mechanical state variable is the force of a myosin motor (Huxley, 1957; Hill, 1974; Pate and Cooke, 1989; Linari et al., 2010; Jarvis et al., 2021; Campbell et al., 2011; Månsson, 2020). These models are based on the obsolete 17th-century philosophy of corpuscular mechanics disproven by Carnot 200 years ago (Baker, 2023a). In contrast, according to a thermodynamic muscle model (Baker and Thomas, 2000a; Baker, 2022b), muscle force is determined from the free energy of a myosin motor ensemble (Eq. 2), where the mechanical state variable is muscle force, $F$.

In 1938, A.V. Hill observed that muscle mechanics, energetics, and kinetics are all functions of muscle force, $F$, implying that $F$ is the mechanical state variable of a thermodynamic muscle system. Based on these observations Hill developed a thermodynamic equation that accurately describes the relationship between muscle force, $F$, muscle shortening velocities, $V$, muscle power output, and muscle heat output (Hill, 1938). In 1999, we observed that the mechanics, kinetics, and energetics of force-generating myosin switches in muscle are functions of muscle force, $F$ (Baker et al., 1999), which is to say that muscle force is determined from the free energy of an ensemble of myosin switches. Based on these observations, we established that the molecular mechanism of muscle contraction is the shortening of an entropic spring consisting of an ensemble of force-generating myosin switches (Baker et al., 1999; Baker, 2022b), where the entropic spring bridges the gap between force-generating myosin switches and A.V. Hill's thermodynamic muscle force (Baker and Thomas, 2000a).

According to thermodynamics, muscle force is mechanically constrained (defined) on only one thermal scale—the thermal scale of muscle. That is, muscle force, $F$, is the only mechanical state variable that can be defined in a model of muscle contraction. At this thermal scale, the stochastic, thermally fluctuating forces of molecules on all smaller thermal scales (including the scale of individual myosin motors) are defined by muscle

force, $F$, not the other way around. In other words, myosin motors function under the constraint of $F$ with their kinetics and energetics defined by $F$ not by molecular forces that corpuscularians imagine to be constrained on the thermal scale of muscle.

Stochastic models are necessary for understanding the emergent behaviors of thermally fluctuating forces on a given thermal scale. Here, we develop a stochastic model of thermally fluctuating forces on the thermal scale of muscle and observe bifurcation between periodic and stochastic dynamic force generation that cannot be observed in mathematical models or continuous computer simulations. The fluctuating structures, degrees of freedom, thermal energy, and entropy defined on the thermal scale of muscle all differ from those defined on smaller scales. For example, the entropic contractile force (Eq. 2) defined here in terms of global protein structures on the thermal scale of muscle differs from entropic forces defined in terms of protein structural components on smaller thermal scales (e.g., the entropic folding forces of proteins). A thermodynamic muscle model implies that a change in thermal scale is a physical, discrete transformation (Baker, 2023b, *Preprint*, 2024a, *Preprint*) and that stochastic simulations cannot span thermal scales but instead must be run as nested simulations of one coarse-grained thermal scale at a time (Aboelkassem et al., 2019), where the simulations presented herein are coarse-grained on the thermal scale of muscle.

Because a thermodynamic model upends our current understanding of how muscle works, additional testing is needed. We show here that four phases of force generation emerge from a simple kinetic simulation of a binding reaction (binding, ergodic, isothermal, and catastrophic), creating a thermodynamic force-generating loop that repeats periodically. The force-generating binding reaction (Eq. 1, binding phase) and isothermal force generation (Eq. 2, isothermal phase) are different thermodynamic phases of force generation that are both well-defined mathematically. If the binding reaction fails to equilibrate (along Eq. 1) with the binding free energy, it equilibrates in a non-ergodic state ($a < 1$) from which an active ergodic phase provides a pathway from Eq. 1 to Eq. 2 (ergodic phase). The phase from Eq. 2 back to Eq. 1 (catastrophic phase) occurs with a catastrophic loss of force.

Muscle must first generate force before it generates power output. For example, in lifting a dumbbell, a force equal to and opposite the weight of the dumbbell must be generated within muscle before power can be transferred to a gravitational potential upon lifting the dumbbell. These are the first two phases of muscle's thermodynamic work loop. We previously described the first phase of muscle's thermodynamic work loop as the binding phase of force generation (Baker, 2022b); however, if power output can be gated, for example, by valves in the heart, this force-generating phase in a work loop may include ergodic and isothermal phases of force generation as well. The second phase of the muscle's thermodynamic work loop is the shortening of the entropic spring (Baker, 2022b), or the thermodynamic power stroke, which occurs with a decrease in force along the binding isotherm (Eq. 2, right-to-left in Fig. 2 A). In other words, a thermodynamic power stroke is the reversal of isothermal force generation (Eq. 2, left-to-right in Fig. 2 A). The

implication is that the isothermal force generated prior to a power stroke increases the size of the subsequent power stroke (e.g., increases the stroke volume in a cardiac cycle). This suggests that the muscle's thermodynamic work loop is optimized by tuning parameters like those varied herein to maximize isothermal force generation prior to a power stroke.

This contrasts with the conundrum posed by a recent corpuscular mechanic characterization of myosin's lever arm rotation as a force-generating power stroke. If actin-induced lever arm rotations generate force, large forces are generated when many motors are in a post-power stroke, $N_{AMD}$, state, or

$$F = N_{AMD} \cdot F_{uni},$$

where $F_{uni}$ is the average force of a bound motor. According to this equation, at large forces, the muscle cannot generate power output because most motors are in a post-power stroke state. This equation also shows that the spontaneous detachment of all myosin motors becomes increasingly improbable at higher forces inconsistent with periodic force generation.

In contrast, in a thermodynamic model, isothermal force generation primes motors for a power stroke and increases the probability of a spontaneous detachment of all myosin motors. According to Eq. 2,

$$F \alpha - \Delta G° + k_B T \cdot \ln(N_{MDP}/N_{AMD}),$$

$N_{AMD}$ decreases with an increase in force because the entropy of the ensemble of switches decreases $[k_B \cdot \ln(N_{MDP}/N_{AMD})]$ with an increase in force (Baker, 2023b, *Preprint*). Therefore, force generation detaches myosin motors, priming them for the subsequent power stroke that occurs with actin–myosin binding along the isotherm in the opposite direction of force generation. Here, the probability of spontaneous detachment of all myosin motors increases with increasing force and is assured when $F$ exceeds that at $N_{AMD} = 1$.

Periodic force-generating loops are observed in in vitro studies of small myosin motor ensembles (Hwang et al., 2021) and in SPOCs in muscle (Fabiato and Fabiato, 1978; Martin et al., 2003), indicating that even though it is not physiological isothermal force generation occurs in these systems. Consistent with the simulations herein, Kaya and colleagues observed force-generating loops in small myosin motor ensembles that bifurcate between periodic and stochastic beating (Hwang et al., 2021; Kaya et al., 2017). In these studies, the effective step size, $d/(aN)$, is observed to decrease with increasing force, consistent with force-dependent ergodicity ($a = F/F_o$). The kinetics, periodicity, stochasticity, and amplitudes of these force-generating loops can be measured in these experiments under different conditions (e.g., ATP, ADP, P concentrations, kinetic rates, optical trap stiffness, numbers of myosin motors, N, etc.). To test our model, these experiments can be compared with simulations like those presented herein. Because very few adjustable parameters are available in this model to compel it to fit data, experiments that differ significantly from the model predictions herein would disprove the model. In other words, the model is highly predictive and thus easily tested.

A binary mechanical model accounts for the apparent discrepancy between the periodic force generation observed by

Kaya and colleagues (Hwang et al., 2021) and the steady-state non-ergodic stall force observed by Lombardi and colleagues (Pertici et al., 2018). Intramolecular forces are non-ergodic and prevent myosin motors from equilibrating with $F_o$, stalling force generation in a frustrated, non-ergodic ($a < 1$) steady state, $F = aF_o$. In the experiments of Kaya and colleagues, intramolecular forces are minimized by flexible S2 domain tethers, enabling ergodicity ($a = 1$) and force-generating loops. In experiments of Lombardi and colleagues, there are no flexible tethers, and intramolecular forces stall force generation in a non-ergodic state, preventing periodic force generation. Kaya observed stalled non-ergodic force generation at low [ATP], consistent with our observations that intramolecular interactions increase at low [ATP] (Stewart et al., 2013). It remains unclear whether isometric muscle force results from a non-ergodic steady-state force or from asynchronous force-generating loops summed over many actin filaments.

Intramolecular forces slow the rate of ADP release, increasing the number of bound motors, which further increases both intramolecular forces and the number of bound motors. These intermolecular forces are thought to contribute to sustained tonic muscle contractions, which implies that tonic muscle force is non-ergodic. A more rapid (phasic) relaxation of muscle force requires fewer bound motors, which is achieved through isothermal force generation.

There are several discrepancies between our simulations and experimental observations. First, in in vitro force assays, relatively long periods with no mechanical activity are observed between force-generating loops. This is easily reproduced in simulations (Kad et al., 2005) by assuming a slower binding rate when all myosin motors are detached and actin filaments are no longer held in close proximity to the surface. With the goal of characterizing periodicities and amplitudes, we did not include this conditional rate in our simulations. Second, in Figs. 6 and 7, we used a relatively slow rate, $v$ (= 50 s$^{-1}$), to study the effects of $N$ and $\kappa_{sys}$ on periodicity and amplitude. A more physiological $v$ (= 200 s$^{-1}$) at $N = 5$ results in a larger number of single binding events as observed experimentally.

Simple kinetic simulations of a force-generating binding reaction account for the muscle force–velocity relationship, the four phases of muscle force transients, the four phases of a muscle work loop, and here the four phases of both stochastic and periodic force-generating loops (Baker, 2022b). The binding equation (Eq. 1) describes phase 2 of a force transient, phases 1 and 3 of a work loop, and phase 1 of isometric force generation. The free energy equation (Eq. 2) describes phase 4 of a force transient, phases 2 and 3 of a work loop, and phase 3 of isometric force generation. It is remarkable that these diverse and complex mechanochemical behaviors all emerge from a single molecular mechanism (Fig. 1 A). This simple binary mechanical model provides a radically new perspective on the mechanisms of muscle and motor ensemble function, which we have been developing and testing for 25 years.

## Acknowledgments

Henk L. Granzier served as editor.

This work was funded by a grant from the National Institutes of Health 1R01HL090938-01.

Author contributions: V. Murthy: Data curation, Formal analysis, Investigation, Methodology, Resources, Software, Visualization, Writing - original draft, J.E. Baker: Conceptualization, Funding acquisition, Methodology, Project administration, Resources, Supervision, Validation, Writing - review & editing.

Disclosures: The authors declare no competing interests exist.

Submitted: 28 September 2023

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

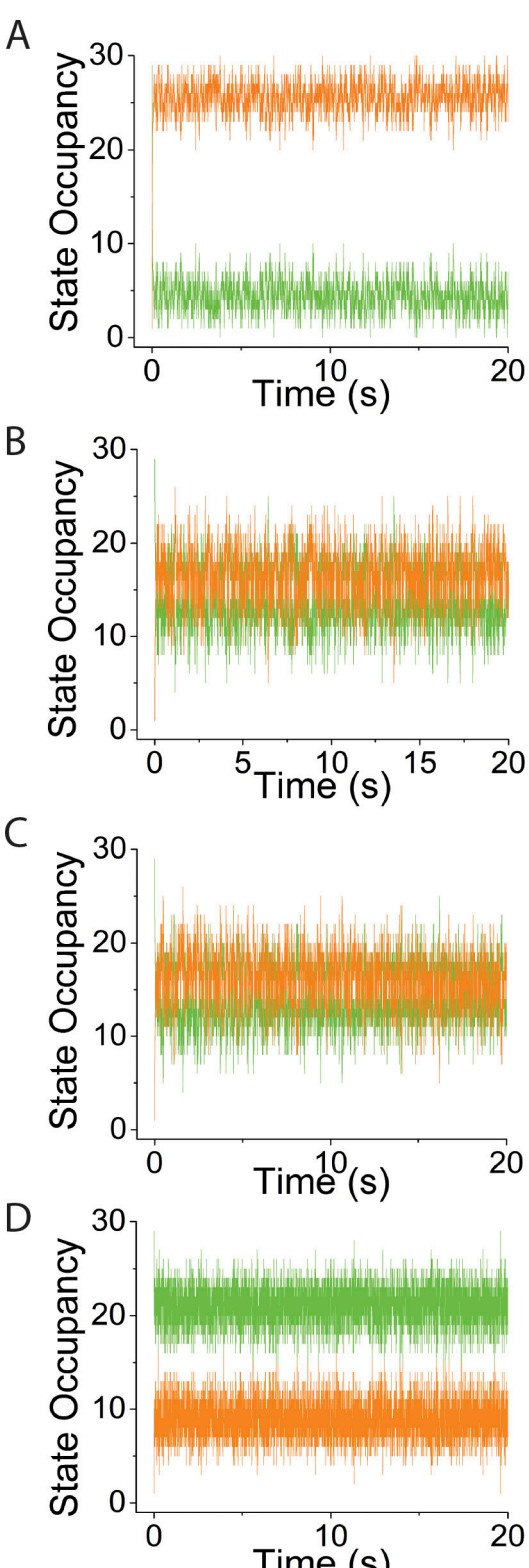

Figure S1.   **Supplement to simulations in** Fig. 3 **varying system spring stiffness, κ_sys, using ν = 0, N = 30. (A–D)** Simulated time courses of $N_{MDP}$ (green trace) and $N_{AMD}$ (orange trace) are plotted at values for κ_sys of (A) 0.01 pN/nm, (B) 0.16 pN/nm, (C) 2 pN/nm, and (D) 10 pN/nm.

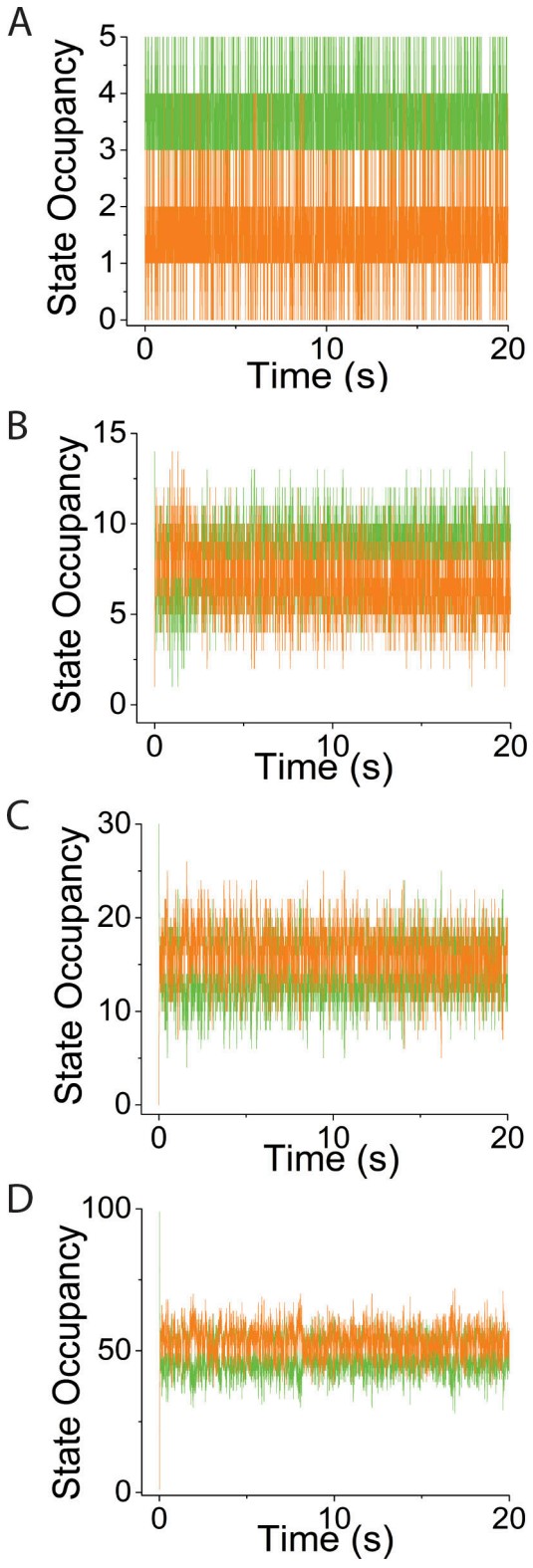

Figure S2.   **Supplement to simulations in** Fig. 4 **varying _N_, using _v_ = 0 and κ_sys = 2 pN/nm. (A–D)** Simulated time courses of $N_{MDP}$ (green trace) and $N_{AMD}$ (orange trace) are plotted at values for _N_ of (A) 5, (B) 15, (C) 30, and (D) 100.

Figure S3. **Supplement to simulations in** Fig. 5, **varying** *v*, **using** *N* = 30 **and** κ_sys = 2 pN/nm. (A–D) Simulated time courses of $N_{MDP}$ (green trace) and $N_{AMD}$ (orange trace) are plotted at values for *v* of (A) 1 s$^{-1}$, (B) 25 s$^{-1}$, (C) 100 s$^{-1}$, and (D) 200 s$^{-1}$.

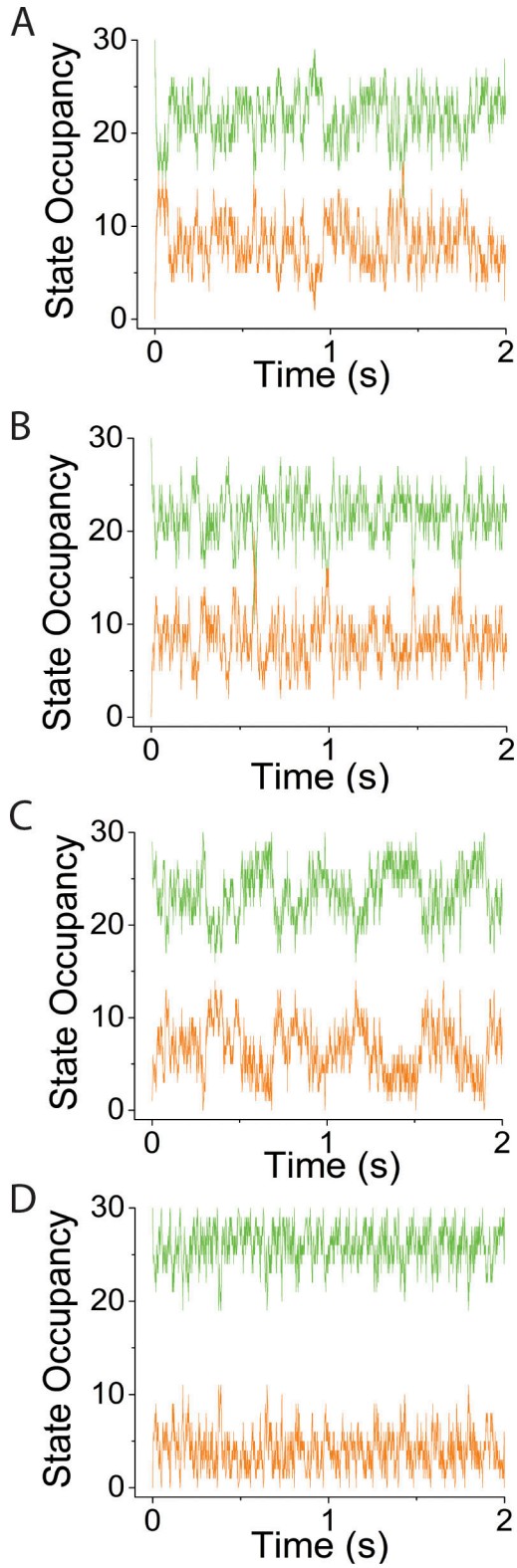

Figure S4. **Supplement to simulations in** Fig. 6 **varying κ$_{sys}$, using $v$ = 50 s$^{-1}$ and $N$ = 30. (A–D)** Simulated time courses of $N_{MDP}$ (green trace) and $N_{AMD}$ (orange trace) are plotted at κ$_{sys}$ values of (A) 0.01 pN/nm, (B) 0.16 pN/nm, (C) 2 pN/nm, and (D) 10 pN/nm.

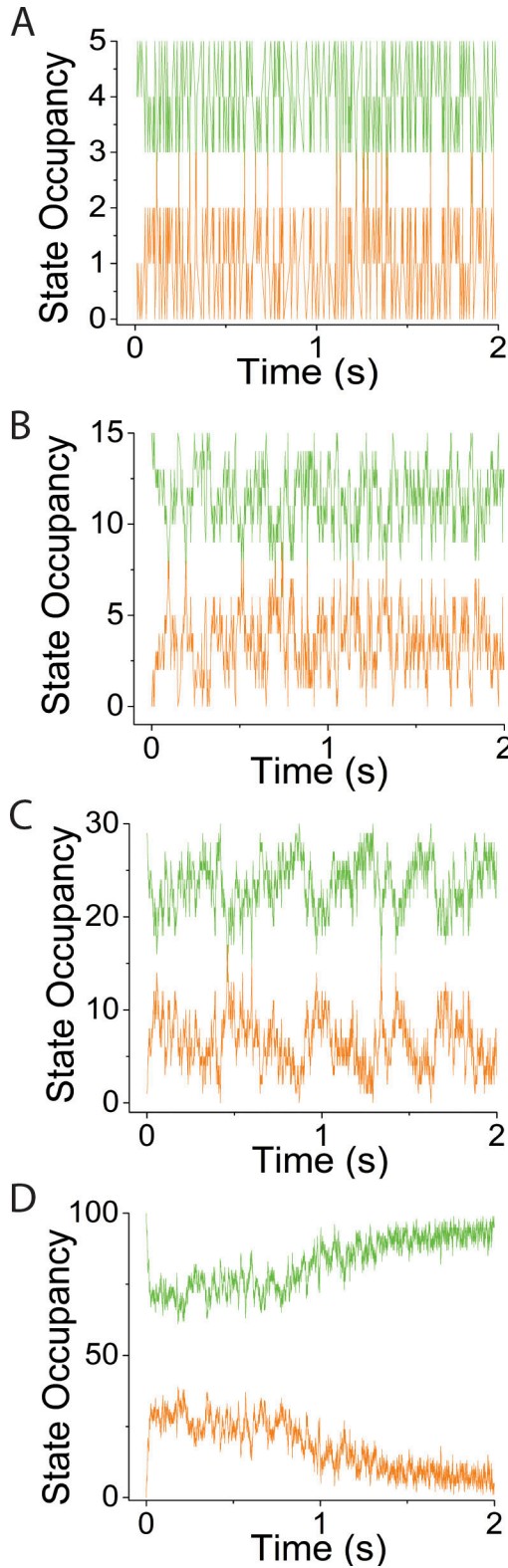

Figure S5. **Supplement to simulations in** Fig. 7 **varying *N*, using *v* = 50 s⁻¹ and κ_sys = 2 pN/nm. (A–D)** Simulated time courses of $N_{MDP}$ (green trace) and $N_{AMD}$ (orange trace) are plotted at $N$ values of (A) 5, (B) 15, (C) 30, and (D) 100.

**Provided online is Data S1. Data S1 provides the sample python code used for simulations.**

