## [Peer Review File · The Journal of General Physiology]

Stochastic Force Generation in an Isometric Binary Mechanical System

Vidya Murthy and Josh Baker

Corresponding Author(s): Josh Baker, University of Nevada, Reno School of Medicine

Review Timeline:

Submission Date:	September 28, 2023
Editorial Decision:	December 12, 2023
Revision Received:	July 2, 2024
Editorial Decision:	July 30, 2024
Revision Received:	August 14, 2024
Editorial Decision:	September 3, 2024
Revision Received:	September 25, 2024

Editor: Henk Granzier

Transaction Report:

DOI: <https://doi.org/10.1085/jgp.202313493>

December 12, 2023

Dr. Josh E Baker
University of Nevada, Reno School of Medicine
Pharmacology
1664 N. Virginia Street, MS 330
Reno, NV 89557

Re: 202313493

Dear Dr. Baker,

Your manuscript entitled "Stochastic Force Generation in an Isometric Binary Mechanical System" has now been seen by 3 reviewers, whose comments are appended below. You will see that the reviewers have raised serious concerns about the manuscript, including the accessibility of your work for a wide audience and many of the statements with regards to earlier work that are unnecessarily strong (as explained in the reviews).

As a result of these concerns, I am afraid that we cannot offer to publish your manuscript. However, if you believe that further work would allow you to address these issues to the satisfaction of the editors and reviewers, we would be willing to consider a substantially revised manuscript if resubmitted within one year. In the case of eventual publication, the article would include a 'revised date' alongside its submitted and accepted dates. In the event that the revision process takes longer, any resubmitted manuscript would be treated as a new submission and would be given a new manuscript number. Either way, any revised manuscript would be sent out for review to the original reviewers, subject to their availability, and we must emphasize that we cannot predict the final outcome.

Should you decide to submit a revised version, please submit your revised manuscript via the link below and include a letter that details your responses to the editors' and reviewers' concerns. Also, to facilitate evaluation of revised manuscript by Reviewers and Editors, please provide a copy of the text with alterations highlighted (e.g., boldfaced, underlined, or in color). If you feel changes are too extensive to provide a version of your paper with changes highlighted, please state so in your response.

Please pay particular attention to recent changes to our instructions to authors in the following sections: Data presentation, Blinding and randomization and Statistical analysis, under Materials and Methods, as shown here: <https://rupress.org/jgp/pages/submission-guidelines#prepare>. Re-review will be contingent on inclusion of the required information (including for data added during revision) and demonstration of the experimental reproducibility of the results. Also, to improve the reproducibility of published content, we have partnered with SciScore. Authors are prompted in eJP to copy and paste the Materials and Methods section of their manuscript for a SciScore assessment when submitting their revised manuscript. Authors are encouraged (not required) to further revise their Materials and Methods if the SciScore is below 4. More information can be found here: <https://rupress.org/jgp/pages/submission-guidelines#sciscore>

Please do not hesitate to contact me (via the editorial office) if you feel that a discussion of the reviewers' and editors' comments would be helpful.

Thank you for the opportunity to consider your manuscript, a copy of which we will retain for our files.

Sincerely,

Henk L. Granzier, Ph.D.
On behalf of Journal of General Physiology

Journal of General Physiology's mission is to publish mechanistic and quantitative molecular and cellular physiology of the highest quality; to provide a best-in-class author experience; and to nurture future generations of independent researchers.

Please submit your revised manuscript via this link within one year:
Link Not Available

Reviewer #1 (Comments to the Authors):

The study from the Baker lab has developed a simple two-state system that expands into a mechanical model to describe

muscle contraction. This builds on their recent work that outlines the fundamental principles and now includes four thermodynamic components that emerge from their simulations.

Overall this paper is important, however there are issues relating to accessibility that need to be addressed. To reach an audience appropriate for this journal it is my opinion that the thermodynamics needs to be more effectively defined and explained. Ergodicity is reasonably well explained, but the remainder of the terms are not immediately clear.

Additional specific comments:

"A.F. Huxley proposed a molecular mechanic model of muscle contraction that violates the principles of thermodynamics"

This is a strong statement and could do with a little more explanation.

"Gibbs described thermodynamics as the mechanical laws of a macroscopic system of particles as measured by an observer, not the mechanical laws of particles conjured by corpuscularians."

This statement felt a little pejorative, I would suggest toning this down.

Could the authors explain in a little more detail the origin of equation 1? This is an important equation and so deserves a more detailed explanation.

Surprisingly, there is no definition of F_0 . And in Fig 2 what is F_0 ?

From paragraph starting: "Figure 5 shows simulations of the model in Fig. 1 introducing irreversible rates..." where is the inset and the panels increase from 1 to 200/s not 100/s.

Reviewer #2 (Comments to the Authors):

The manuscript of Murthy and Baker use a thermodynamic model based on a two-state kinetic scheme to simulate muscle force generation in isometric conditions. Under the set of hypotheses, the simulations show a generation of force divided in four different phases with different thermodynamic characterizations. The results are associated to the experimental data present in the literature about small myosin motor ensembles and spontaneous oscillatory contractions.

A better understanding of the link between mathematical modeling of muscle mechanics and the underlying thermodynamics is surely needed in the current state of the art, as it is a known and recognized limit in many published mathematical models.

However, the manuscript presents several critical issues to be proposed as a "Hypothesis" which aims is to "report novel theoretical analyses or interpretations of existing data that help to define a new perspective for future investigations".

It is not clear what are the improvements of the approach proposed in the manuscript respect the published models which are somehow derived from the seminal paper of A.F. Huxley in 1954. Despite the known limits in the thermodynamic interpretation, these models have been used by several authors to shed lights on several mechanisms in actomyosin interaction, as for example the Huxley and Simmons 1971 model. This approach also emphasized the limits of previous models, as, just to stay on the historical examples, the Huxley Tideswell Model in 1996, prompting more experimental data and new models. The needs of making these kind of models more complicated than simple binary systems, proceeded in strict correlation with the most up-to-date structural evidences from X-ray data, fluorescent probes and cryo-EM analyses (among other techniques), to better match the real nature of muscle mechanics. For these reasons, to confine these kind of models into the corpuscular mechanic philosophy, almost with no explanation, is unacceptable in my opinion, but apart from this, to be an "hypothesis" the manuscript should prove the actual improvements of the new approach, which is not the case. At the end of the Discussion section it is said that the binary mechanical model accurately accounts for muscle chemistry and mechanics but it is not shown in the present manuscript. As far as I can understand from the manuscript, it is a presentation of numerical simulations based on an already published model, to qualitatively interpret the behavior of few motors ensemble under some hypothesis.

In the introduction, it is said that the model is used to perform simulations over a wide range of conditions but the results are more a parametric analysis as recognized also in the discussion. While interesting, it is not a way to define a new perspective for future investigation, since in the physiological contraction of muscle these parameters are not changing. It may be interesting for single or few motors experiments, as indicated in the text, but it is not clear how it can prompt a new perspective.

The manuscript proposes a numerical simulation of an already published model (ref. 18), however, the description of the model proposed in the manuscript is too short and extremely hard to follow. Comparing the description with the much longer one proposed in ref. 18, I can deduce that there are also some errors (a sign in eq. 2, an inequality in the first page of the results), also some definitions seems to be missing or in a wrong position (f^0 are defined as unloaded rate constants in ref 18 while here they are defined just as forward and reverse constants, N is defined several lines after its first appearance), it is not explained the choice of using 0.5 in splitting eq. 3 in two different equations, also comparing Eq. 3 with eq. 7 in ref 18 it seems that there is one more term. The derivation of the term N_{AMD+1} in Δ_rG is not properly described as well as why it is then approximated just as N_{AMD} . The relationship between the force and the amount of "AMD" should be made more clear given the strong discrepancy between their temporal development shown in Fig. 2B and 2C. The parameter "a" and the needs of using the term $d/(a/N)$ is not well explained.

Minor issues:

Acronyms MDP and AMD should be defined

Beginning of results: plot against time should be specified, also, reference to Fig 2B and Fig 2C seems to be switched
"v=0 (Figs. 5E and 5F) should be 5 A and 5B I think

Second line of Discussion, "Eq. 1" or "Fig.1"

"Often erroneously referred to as a "duty ratio": it should be rephrased since many definitions of duty ratio are actually correct in the framework of the papers which present it in that way. I suggest to avoid such a general critic, but it should be made much more explicit if it is really needed.

Reviewer #3 (Comments to the Authors):

JGP #202313493

Stochastic Force Generation in an Isometric Binary Mechanical System

NB: PDF file was sent to the editor as well.

The authors have presented modeling and simulations of a reversible two-state binding reaction that alters a system spring, incorporating forward and reverse rates derived from the binding free energy. This single force-generating mechanism gives rise to four distinct phases of force generation: adiabatic, isentropic, ergodic, and catastrophic. Among these, two phases conform to ideal thermodynamic processes. Adiabatic (isometric) force generation is elucidated by the force equation, while isothermal force generation is expounded by the binding free energy. These ideal processes establish benchmark equations against which all other non-equilibrium, non-ideal force-generating processes are assessed.

While the model captures valuable information and data concerning the stochastic force generation in an isometric binary mechanical contraction system, I have a few points that require clarification.

Major Points

Could the authors emphasize the distinctions between the work presented here and their earlier publication (Ref. 18)? I comprehend that the current investigation successfully incorporates the stochastic mechanics of myosin motor ensembles, whereas the prior model (Ref. 18) exclusively represents smooth trajectories. However, there is a lack of clarity regarding how stochasticity was introduced in the simulations, particularly given that both models employ the same mathematical framework. It would be beneficial if the authors could provide insights into the specific methodological variances between the two studies (the present one and Ref. 18).

It appears that the smooth trajectory approach outlined in equation (18) effectively delineates the force generation phases. In contrast, the current methodology lacks a distinct transition or bifurcation between phases, particularly evident when plotting the (F-NMPD) relationship in the majority of simulated cases. Could you clarify the numerical criteria used to identify transition points between these force generation phases?

Could you please offer additional information regarding the simulation protocol and the numerical algorithm employed to produce the results presented?

Minor Points

Methods section: (Please review all the equations)

There is a typo (missing a minus sign) in equation (2) which should read as
$$F = -aN[\Delta G_0 + k_B T \ln(NAMD/NMDP)]/d$$

There is a typo. The temperature T is missing in the following expressions i.e, KB should be replaced by KBT. The two equations after equation 3.

Results section

"From simulations of a two-state binding reaction (Fig. 1), we plot time traces of F (Fig. 2B) and the occupancies of the AMD (NAMD, orange line) and MDP (NMDP, green line) states (Fig. 2C).' It appears that there is a misplacement in the citation order of Figures B and C. The traces for F are displayed in Figure C, whereas the occupancies are presented in Figure B.

In Figure 2A, I suggest incorporating an arrow to denote the "Catastrophic" phase.

In figure 3, what is the rationale behind choosing these particular values of the spring constants $k_{sys} = 0.01, 0.16, 2$ and 10 pN/nm. Please add a reference if possible.

In figures (3-7), it would be beneficial to display plots illustrating the state occupancies over time for all simulated cases, akin to the format shown in Figure 2B.

I highly suggest reconsidering the plotting of figures (3-7) by improving the scaling and opting for higher resolution. Adjusting the Y-axis limit would enhance the visibility of the data. Similar adjustments could be beneficial for the inset "a" figures as well.

Detecting the presence of the four loops in force generation can be challenging in certain scenarios. The transition or onset from one phase to another appears to be highly responsive to the range of parameters. It would be highly beneficial if you could share the simulation code, preferably in Matlab. Providing such code would greatly assist readers in gaining a clearer comprehension of the intricacies involved in the four loops of force generation.

Re: 202313493

Dear Dr. Baker,

Your manuscript entitled "Stochastic Force Generation in an Isometric Binary Mechanical System" has now been seen by 3 reviewers, whose comments are appended below. You will see that the reviewers have raised serious concerns about the manuscript, including the accessibility of your work for a wide audience and many of the statements with regards to earlier work that are unnecessarily strong (as explained in the reviews).

I thank the editor and three reviewers for their careful reading and constructive critiques of our manuscript. I believe that we were able to adequately address all reviewer concerns in the revised manuscript. These revisions were extensive spanning all sections of the manuscript and are therefore not marked in the revised manuscript. We appreciate that the reviewers identified several typos, which we have corrected in the revised manuscript. The reviewers also point out clear weaknesses in the description of our stochastic computational model, which we greatly improve in our revised manuscript. We address additional useful reviewer recommendations below and in the revised manuscript. These revisions have significantly improved the manuscript and for that we are grateful.

Serious concerns about the manuscript warrant a comprehensive response. The editor lists "accessibility" and "unnecessarily strong" as the two major concerns.

Accessibility. We continue working to make the physics of muscle contraction more accessible to a general readership. Over a period of 25 years, I have established a substantial body of research developing and testing a thermodynamics muscle model [see (1)], and over the past year I have written several additional manuscripts that further describe the model and its implications. These manuscripts are either published or in review and available on the BioRxiv server. For a broad perspective, "The Problem with Inventing Molecular Mechanism to Fit Thermodynamic Equations" (1) is a recently published commentary that provides an historical perspective describing how corpuscular mechanic models (models that assume the mechanics of a system are described by the mechanics of individual molecules within that system) like Huxley's "springs of myosin" and Boyle's "springs of air" were disproven by Carnot in 1824. It summarizes 25 years of experimental support for a thermodynamic muscle model. It also contains a table that summarizes the fundamental differences between corpuscular mechanic and thermodynamic interpretations of key mechanisms of muscle contraction and the experiments that challenge the former and support the latter.

"The Kinetic and Energetic Pull of Chemical Entropy" (2) develops in greater detail the kinetics and energetics presented here and originally in my Langmuir paper. "Four Phases of a Muscle Force Transient Emerge from a Binary Mechanical System" (3) demonstrates that the same two-state thermodynamic model that here predicts four phases of isometric force generation accurately describes the four phases of force transients observed following rapid chemical and mechanical perturbations of isometric muscle. "A Macroscopic Quantum Heat Engine: Nested Resonant Structures and the Entropic Stability of Life" (4) describes the implications of muscle thermodynamics for thermal scaling. "Mixed Scale Versus Multiscale Models of Muscle Contraction" is a letter to the editor challenging a corpuscular mechanic model proposed recently in Biophysical Journal. I will be relentless in writing these letters until someone

formally refutes that thermodynamics disproved corpuscular mechanics 200 years ago (the entropic force of an ensemble of molecules is not defined by the forces of individual molecules).

There are also manuscripts in the final stages of preparation that I can make available to reviewers. “What is the Power Stroke Mechanism of Muscle Contraction?” provides an historical perspective on how the power stroke mechanisms of muscle contraction has changed over the years and remains to this day undetermined, where currently a “power stroke” is widely defined as a structural change that is not a stroke and that does not generate power output. “The Mechanical Autoinhibition of a Macromolecular Assembly” is a thermodynamic description of how the force generated by muscle inhibits ADP release, not on the scale of an individual myosin motor, but on the scale of an ensemble of myosin motors. In that manuscript, we present experimental data in support of this hypothesis. “The Chemistry of Muscle Force” is an experimental study that demonstrates a strong correlation between the actin-myosin binding free energy and muscle force as predicted by a thermodynamic muscle model. I am currently working on a perspective piece “The Physics of Muscle: How Thermodynamics Forged a Multi-Scale Binding Engine” that summarizes much of this work. I also recommend the very accessible textbook that I frequently reference, Thermal Physics by Kittel and Kroemer, which describes the thermodynamics and statistical mechanics of a binary spin system analogous to a binary mechanical system.

In short, we continue to establish a strong foundation for the thermodynamics of muscle. The challenge with any one of our manuscripts, however, is making muscle thermodynamics accessible to a broad readership within a single article. This challenge is amplified for a readership that ignores our work and that on average has never studied thermodynamics. This challenge is further amplified for a readership that is currently captivated by a structural determinism (e.g., a molecular “power stroke” defined as a structural change) that is the antithesis of thermodynamics.

After 25 years I remain undaunted by these challenges and greatly appreciate guidance from reviewers on how to overcome them. The implications of thermodynamics for biological systems are profound and remain largely unexplored on the thermal scale of biological systems, which is why I am passionate about teaching thermodynamics to biologists, and why we strive to make the thermodynamics in our manuscripts as accessible as possible. In our revised manuscript, we provide a broader more conceptual description of the model, which we are hopeful will make it more accessible. We also more heavily reference detailed derivations presented in other papers.

In the end, the model is really not that complicated. It is nothing more than two simple equations. Equation 1 is a Hookean spring equation that states that muscle force, F , increases with the number of actin-bound myosin motors, N_{AMD} . This relationship is non-linear when force-generating step sizes are force-dependent (i.e., non-ergodic). Equation 2 is the Gibbs free energy equation for actin-myosin binding, which defines the F -dependence of the forward and reverse binding rates. I am unaware of any other model that can be summarized in its entirety in three sentences while accounting for most key aspects of muscle contraction (the force-velocity relationship, the four phases of a tension transient, a thermodynamic work loop, etc). Equations 1 and 2 are mathematical solutions from which Hill’s force-velocity relationship is derived. The derivative of Eq. 1 with respect to time provides either a continuous computational solution (by solving the differential equation) or a stochastic computational solution (through Monte Carlo time steps). The simulations in Langmuir (5) are the former and the simulations here are the latter.

Unnecessarily Strong Statements about Earlier Work. Corpuscular mechanics assumes that a system force is fully defined by the forces of individual molecules within that system (e.g., Boyle's 17th century "springs of air" and Huxley's 1957 "springs of myosin"). In contrast, thermodynamics assumes that a system force is entropic and cannot be defined by the forces of individual molecules within that system. Clearly, these are mutually exclusive theories. According to thermodynamics, Huxley-like models are wrong, and according to Huxley-like models, thermodynamics is wrong. It is unclear why **scientific** statements based on the former are here considered offensive and raise serious concerns, while **conspiratorial** statements (see Reviewer 2 comments) based on the latter are accepted.

Thermodynamics disproved corpuscular mechanics 200 years ago (i.e., an entropic system force cannot be accounted for by the forces of individual molecules in that system). With absolutely no scientific justification, reviewer 2 reverses this well-established scientific argument for the sole purpose of defending a scientific community's belief in an obsolete 17th century philosophy (corpuscular mechanics). This is not science. It is a conspiratorial effort to reverse one of the most important advances in the history of science. In other words, Reviewer 2's argument is a corpuscularian conspiracy theory that has been used by a scientific community for more than half a century to reject thermodynamics. The social psychology underlying this historic scientific blunder will be scrutinized for generations. Unlike in this review, history will not hold me to account for offending this scientific community with scientific statements or for trying to make thermodynamics more accessible to them. History will hold this scientific community to account for being offended by scientific statements and for continuing to reject thermodynamics 25 years after the thermodynamic mechanism of muscle contraction was discovered (the shortening of an entropic spring consisting of an ensemble of force-generating myosin motor switches).

These strong scientific statements have remained formally unchallenged for a quarter century. If there was a scientist among them, corpuscularians would long ago have translated their defensive responses to these statements into formal scientific arguments. Unable to do so, a scientist would conclude that these are legitimate scientific statements in need of further testing, while a conspiracy theorist is offended by legitimate scientific statements and rejects them with arguments that are limited to their conspiratorial convictions (see reviewer 2's arguments – the latest of many reviews by corpuscularians over the past quarter century that resemble the arguments of 16th century geocentrists). My hope is that this review can move past true believers defending their beliefs and can focus instead on the physics (thermodynamics) of muscle. I am grateful to reviewers 1 and 3 for doing just that.

Because Huxley-like models are irrelevant to a thermodynamic model, we strongly support minimizing formal ("strong") comparisons to Huxley-like models. Likewise, because Huxley-like models are irrelevant to a thermodynamic model, Reviewer 2 should refrain from challenging a thermodynamic model on the basis that it is not a Huxley-like model. If we can move past Huxley-like models, then we agree that scientific statements that offend corpuscularians are unnecessary, and in our revised manuscript we tone them down considerably.

Reviewer #1:

Overall this paper is important, however there are issues relating to accessibility that need to be addressed. To reach an audience appropriate for this journal it is my opinion that the thermodynamics needs to be more effectively defined and explained. Ergodicity is reasonably well explained, but the remainder of the terms are not immediately clear.

We appreciate Reviewer 1's recognition of the importance of this paper. In our revised manuscript, we clean up our descriptions of the two key equations. Equation 1 is simply a spring equation. Equation 2 is simply a Gibbs binding free energy equation. In our revised manuscript, we improve descriptions of these equations as well as descriptions of the step size and rate equations. We also more heavily reference where readers can find more detailed derivations/descriptions.

Additional specific comments:

"A.F. Huxley proposed a molecular mechanic model of muscle contraction that violates the principles of thermodynamics"

This is a strong statement and could do with a little more explanation.

In our revised manuscript, we expand upon this.

Briefly, it is physically impossible to describe the entropic force of an ensemble of molecules (a thermodynamic model) in terms of the forces of the individual molecules in that ensemble (Huxley-like corpuscular mechanic models). This was established in 1824 by Sadi Carnot who showed that system entropy disproves corpuscular mechanic models like Boyle's description of gas pressure as the sum of forces of the "springs of air" and Huxley's description of muscle force as the sum of forces of the "springs of myosin". Carnot's arguments (that systems of molecules contain entropy) is the foundation of the second law of thermodynamics. In other words, like Boyle's "springs of air" Huxley 1957 violates the second law of thermodynamics because "springs of myosin" do not account for the entropic force of a myosin motor ensemble.

Consistent with the second law of thermodynamics, we have shown that the mechanism of muscle contraction is not a molecular power stroke (the shortening of a molecular spring); it is an ensemble power stroke (the shortening of an entropic spring consisting of an ensemble of force-generating myosin motor switches). We have been developing and testing this thermodynamic model of muscle contraction for the past 25 years, as summarized in a recent commentary (1).

"Gibbs described thermodynamics as the mechanical laws of a macroscopic system of particles as measured by an observer, not the mechanical laws of particles conjured by corpuscularians."

This statement felt a little pejorative, I would suggest toning this down.

In our revised manuscript, we remove this statement.

Could the authors explain in a little more detail the origin of equation 1? This is an important equation and so deserves a more detailed explanation.

In the revised manuscript, we improve our description of Eq. 1.

Surprisingly, there is no definition of F_0 . And in Fig 2 what is F_0 ?

F_0 is the force along the equilibrium binding isotherm. In the revised manuscript, we define this more clearly.

From paragraph starting: "Figure 5 shows simulations of the model in Fig. 1 introducing irreversible rates..." where is the inset and the panels increase from 1 to 200/s not 100/s.

We did not include insets in Figs. 5 and 6 because of limited space and because at some point plotting ergodicity, a , over time becomes redundant. We have removed references to insets in the Fig. 5 and 6 legends. We have also changed 100 to 200. Thank you for pointing out these typos!

Reviewer #2:

A better understanding of the link between mathematical modeling of muscle mechanics and the underlying thermodynamics is surely needed in the current state of the art, as it is a known and recognized limit in many published mathematical models. However, the manuscript presents several critical issues to be proposed as a "Hypothesis" which aims is to "report novel theoretical analyses or interpretations of existing data that help to define a new perspective for future investigations".

It is not clear what are the improvements of the approach proposed in the manuscript respect the published models which are somehow derived from the seminal paper of A.F. Huxley in 1954. Despite the known limits in the thermodynamic interpretation, these models have been used by several authors to shed lights on several mechanisms in actomyosin interaction, as for example the Huxley and Simmons 1971 model. This approach also emphasized the limits of previous models, as, just to stay on the historical examples, the Huxley Tideswell Model in 1996, prompting more experimental data and new models. The needs of making these kind of models more complicated than simple binary systems, proceeded in strict correlation with the most up-to-date structural evidences from X-ray data, fluorescent probes and cryo-EM analyses (among other techniques), to better match the real nature of muscle mechanics. For these reasons, to confine these kind of models into the corpuscular mechanic philosophy, almost with no explanation, is unacceptable in my opinion, but apart from this, to be an "hypothesis" the manuscript should prove the actual improvements of the new approach, which is not the case.

Corpuscular mechanics assumes that the mechanics of individual molecules in a system account for the mechanics of that system. We do not confine Huxley-like models to the corpuscular mechanic philosophy. This is done by those who assign springs to myosin motors and assume that molecular power strokes (the work performed by shortening molecular springs) directly account for muscle's power stroke (the work performed by shortening muscle). In contrast, a thermodynamic model assigns an entropic spring to an ensemble of myosin motors and assumes that an ensemble power stroke (the shortening of an entropic spring) accounts for muscle's power stroke (the work performed by shortening muscle). These two models are mutually exclusive because it is physically impossible to define the

entropic force of an ensemble of molecules in terms of springs assigned to individual molecules in that ensemble. Clearly, a thermodynamic model fits reviewer 2's quoted definition of "hypothesis". It is a novel theoretical analysis. It is a novel interpretation of existing data (four phases of force generation do not emerge from a two-state Huxley 1957 model). And it defines a new perspective. Nowhere in this definition, however, does it state that a new hypothesis must "prove actual improvements" over existing hypotheses. This was an argument used by geocentrists in the 16th century to challenge heliocentrism, and it is an argument used by corpuscularians today to challenge thermodynamics.

Given two competing hypotheses, the task for scientists is to disprove one or the other, and corpuscular mechanic models like Huxley 1957 were disproven 200 years ago. In 1824 Sadi Carnot showed that the entropy of an ensemble of molecules disproves corpuscular mechanic models like Boyle's 17th century "springs of air" (and preemptively Huxley's "springs of myosin") because changes in the entropic energy and force of an ensemble of molecules cannot be accounted for by changes in the energy and force of individual molecules in that ensemble.

Over the past 25 years, I have established a substantial body of work that further disproves corpuscular mechanic models of muscle like Huxley 1957. In a recent commentary (1), I summarize this 25 year effort and describe the difference between thermodynamics and corpuscularianism, the history of these theories, why thermodynamics disproves corpuscularianism, the differences between thermodynamic and corpuscularian interpretations of key aspects of muscle contraction, and the data that support the former and challenge the latter.

In contrast, for 200 years Carnot's argument (i.e., the foundation of the second law of thermodynamics) has gone unchallenged. For 85 years, A.V. Hill's thermodynamic model of muscle contraction has gone unchallenged. And for a quarter century, our proposal that the chemical basis for a thermodynamic muscle model is an entropic spring consisting of an ensemble of force-generating myosin motor switches has gone unchallenged. In fact, A.V. Hill's thermodynamic model and our proposed molecular mechanism continue to gain experimental support.

Because the preponderance of evidence supports a thermodynamic muscle model over corpuscular mechanic muscle models, the burden of proof lies with corpuscularians not thermodynamicists. Because Huxley 1957 was disproven in 1824, Huxley in 1957 needed to disprove the second law of thermodynamics (i.e., demonstrate that there is no entropic muscle force), which – despite protests from A.V. Hill – he never did. It then became incumbent upon every scientist who interprets data using a molecular power stroke model to disprove the second law of thermodynamics. No one ever has. Accounting for new observations by inventing new corpuscular mechanisms demonstrates the obvious: that unconstrained by the second law of thermodynamics we can invent corpuscular mechanisms to describe anything. This is precisely what geocentrists did in the 16th century to "disprove" heliocentrism before Newton's laws disproved unconstrained geocentric mechanisms. The "success" of this purely creative exercise is Kuhn's (6) fallacy of the usefulness of invented mechanisms. Inventing unconstrained corpuscular mechanisms does not justify corpuscular mechanics, and it certainly does not disprove the second law of thermodynamics.

Muscle models have undergone an unprecedented 85-year scientific regression from 19th century thermodynamics (A.V. Hill, 1938) to 17th century corpuscular mechanics (Huxley, 1957) to a structural determinism widely invoked today that predates the scientific revolution (i.e., form defines function with a structural change defined as a "power stroke"). A thermodynamic model of muscle contraction

rectifies this historic scientific blunder. Yet, for the past quarter century, countless scientists have rejected muscle thermodynamics based on a defense of corpuscularianism. It is easy to imagine how observations of protein structures misled scientists into revisiting corpuscular mechanics. However, when these scientists became true believers in this obsolete 17th century worldview and used it to prevent a return to post-19th century science, they joined the ranks of conspiracy theorists, rejecting physical laws in defense of their corpuscularian convictions.

The answer to whether or not thermodynamics applies to muscle has existed for 700 million years. To dismiss the possibility that muscle evolved constrained by the laws of thermodynamics is unscientific and misguided. The best we can do as scientists is to disprove a thermodynamic muscle model, which no one has ever done and, I am quite certain, no educated scientist will ever attempt to do. This is certainly not the case for corpuscular mechanic models of muscle [see (1)]. While the many scientists who have rejected the possibility that thermodynamics applies to muscle may consider their actions a noble defense of a revered legacy paradigm, their actions have in fact left a dark stain on the history of science. I implore reviewer 2 to not contribute to this history with yet another ugly rejection of thermodynamics in defense of corpuscularianism. If reviewer 2's objections to thermodynamics are in fact formally well developed, publishing them would help balance a very lopsided scientific debate. In the meantime, here the task at hand is to objectively assess the predictions of a thermodynamic muscle model. Invoking Huxley-like models, their assumptions, predictions, requirements, history, popularity, or unconstrained capacity to describe anything are all defenses of corpuscularianism and are entirely irrelevant to the assessment of the thermodynamic model at hand.

Reviewer 2 states that there was a "need" to make models more complicated than a binary system. To imply that this corpuscularian requirement extends to thermodynamics is a defense of corpuscularianism that rejects thermodynamics as a possible alternative hypothesis. Complications needed by corpuscularian models are clearly not needed in a thermodynamic muscle model. In a thermodynamic model, a force-generating myosin switch induced by actin binding and gated by Pi release is sufficient to account for muscle's force-velocity relationship, the four phases of a tension transient, a thermodynamic work loop, etc. This is precisely the uncomplicated molecular mechanism of muscle contraction that is directly observed both in muscle (7, 8) and in single molecule mechanics studies (9). Of course, we must continue to test a thermodynamic model to determine if there are observations that require more than two states (a thermodynamic model does not preclude additional states), which is the goal of this and other manuscripts published and in preparation.

At the end of the Discussion section it is said that the binary mechanical model accurately accounts for muscle chemistry and mechanics but it is not shown in the present manuscript. As far as I can understand from the manuscript, it is a presentation of numerical simulations based on an already published model, to qualitatively interpret the behavior of few motors ensemble under some hypothesis.

In the revised manuscript, we make it clear that a binary mechanical model has been shown to accurately account for the muscle force-velocity relationship in (5, 10) and accurately account for the four phases of a muscle force transient following both chemical and mechanical perturbations of isometric muscle in (3). "Already published" in Langmuir are qualitative model descriptions of these and other aspects of muscle contraction. By "already published", reviewer 2 clearly does not mean that we successfully demonstrated in Langmuir (5) that a thermodynamic model accurately describes muscle

contraction. On the contrary, by “already published” reviewer 2 means that we should not be allowed to test whether a thermodynamic model accurately describes muscle contraction. Of course, this is yet another egregious defense of an obsolete legacy paradigm and is incredibly hypocritical considering that a corpuscular mechanic model “already published” in 1957 did not preclude hundreds of corpuscular mechanic models from being published over the following 65 years.

In the current manuscript, we develop the novel, explicit, quantitative prediction that there are four phases of isometric force generation and we systematically establish the predicted effects of Pi, ADP, ATP, stiffness, numbers of myosin motors, binding free energy, step size, ergodicity, etc. on each of these four phases. This is precisely how model testing is accomplished. Perhaps this is unfamiliar because corpuscular mechanic model development is more reactive than predictive. That is, when a new observation is made, a new corpuscular mechanism is simply invented to account for that observation. A model that has this unconstrained flexibility to explain anything has the capacity to predict nothing. A thermodynamic model is constrained by the laws of thermodynamics. It is constrained by a single mechanical mechanism observed in both muscle and single molecules. These constraints make it highly predictive of novel, explicit, quantitative, testable, and often unexpected mechanical behaviors of muscle. This is all minimized by reviewer 2 who outrageously dismisses thermodynamics as “some hypothesis”. Again, dismissing thermodynamics is not a challenge to thermodynamics; it is a passive-aggressive defense of an obsolete legacy paradigm.

In the introduction, it is said that the model is used to perform simulations over a wide range of conditions but the results are more a parametric analysis as recognized also in the discussion. While interesting, it is not a way to define a new perspective for future investigation, since in the physiological contraction of muscle these parameters are not changing. It may be interesting for single or few motors experiments, as indicated in the text, but it is not clear how it can prompt a new perspective.

In the revised manuscript, we now refer to varying parameters as a parametric analysis (not conditions).

Working backward, our simulations do not prompt a new perspective. A thermodynamic model of muscle contraction was prompted by the laws of thermodynamics, the work of A.V. Hill (11), and our observation 25 years ago that muscle is an entropic spring consisting of an ensemble of force-generating myosin motor switches (7, 8, 10). This model fundamentally changes our understanding of how muscle works, and so it is important to test it by establishing novel, explicit, testable predictions. The four phases of isometric force generation that emerge from this model are novel predictions unique to a binary thermodynamic model, and in this manuscript we establish how stiffness, numbers of motors, chemical kinetics, binding free energy, ergodicity, and myosin motor step size affect each of these phases. These are explicit predictions that can be tested experimentally to disprove or support a thermodynamic model of force generation.

In our revised manuscript, we expand upon our discussion of how these four phases of muscle force generation can be interpreted within the context of muscle contraction. One key issue discussed is whether simulated force generating loops average over many filaments to create a constant isometric force or whether a constant isometric force results from non-ergodic force generation. We discuss how and why this might differ among different muscle types (e.g., phasic vs. tonic). It is absolutely not true that these parameters are all static under physiological conditions. Many of these parameters change with muscle remodeling, post-translational modifications, fibrosis, small molecule therapeutics,

mutations, regulation, etc. Our parametric analysis provides a new perspective on how these changes affect muscle physiology.

The manuscript proposes a numerical simulation of an already published model (ref. 18), however, the description of the model proposed in the manuscript is too short and extremely hard to follow.

In our revised manuscript, we significantly improve the description of our stochastic computational protocol. We improve the conceptual description of the model and more heavily reference more explicit derivations. We also reference four new manuscripts that further describe and develop this model and its implications for chemical kinetics and thermal scaling. "The kinetic and energetic pull of chemical entropy" expands upon the kinetic and energetic analysis presented in the Langmuir paper.

Comparing the description with the much longer one proposed in ref. 18, I can deduce that there are also some errors (a sign in eq. 2, an inequality in the first page of the results),

We are grateful to reviewer 2 for catching these typos, and we have corrected them in the revised manuscript.

also some definitions seems to be missing or in a wrong position (f^0 are defined as unloaded rate constants in ref 18 while here they are defined just as forward and reverse constants, N is defined several lines after its first appearance), it is not explained the choice of using 0.5 in splitting eq. 3 in two different equations, also comparing Eq. 3 with eq. 7 in ref 18 it seems that there is one more term.

Regrettably, we use a different nomenclature here than what we used in the Langmuir paper (5). The nomenclature used here aligns with a manuscript under review (2) that develops in greater detail the kinetics and energetics presented in the Langmuir paper. The reason for this change in nomenclature is that rate constants, f^0 , correspond to standard reaction free energies, ΔG^0 , and we wanted the superscript naught to be consistent. Specifically, here and in (2) we define f_+ and f_- as rates and f_+^0 and f_-^0 as rate constants, whereas in Langmuir we defined f_+ and f_- as rate constants. This accounts for the additional term in the exponent in Eq. 3. In our revised manuscript, we make this more clear.

The derivation of the term $N_{AMD} + 1$ in Δ_rG is not properly described as well as why it is then approximated just as N_{AMD} . The relationship between the force and the amount of "AMD" should be made more clear given the strong discrepancy between their temporal development shown in Fig. 2B and 2C.

The term containing $N_{AMD} + 1$ was derived in (5) and more fully in (2). It is the change in entropic energy associated with a chemical step. In the revised manuscript we describe its formal origins and more clearly reference where the derivation can be found. In the revised manuscript, we also state that $N_{AMD} + 1$ is approximately N_{AMD} for large N_{AMD} . In the revised manuscript, we now more clearly describe the relationship between force and N_{AMD} and why Eqs. 1 and 2 describe opposite relationships.

The parameter "a" and the needs of using the term $d/(aN)$ is not well explained.

In the revised manuscript, we improve the discussion of the parameter a and the displacement $d/(aN)$ and more clearly reference the original derivations.

Minor issues:

Acronyms MDP and AMD should be defined

Done.

Beginning of results: plot against time should be specified, also, reference to Fig 2B and Fig 2C seems to be switched

We are grateful to reviewer 2 for catching this typo, and we fixed it in the revised manuscript.

"v=0 (Figs. 5E and 5F) should be 5 A and 5B I think

We are grateful to reviewer 2 for catching this typo, and we fixed it in the revised manuscript.

Second line of Discussion, "Eq. 1" or "Fig.1"

We have fixed this in the revised manuscript.

"Often erroneously referred to as a "duty ratio": it should be rephrased since many definitions of duty ratio are actually correct in the framework of the papers which present it in that way. I suggest to avoid such a general critic, but it should be made much more explicit if it is really needed.

We follow reviewer 2's advice and remove this discussion in the revised manuscript.

Reviewer #3:

JGP #202313493

Stochastic Force Generation in an Isometric Binary Mechanical System

NB: PDF file was sent to the editor as well.

While the model captures valuable information and data concerning the stochastic force generation in an isometric binary mechanical contraction system, I have a few points that require clarification.

Major Points

Could the authors emphasize the distinctions between the work presented here and their earlier publication (Ref. 18)? I comprehend that the current investigation successfully incorporates the stochastic mechanics of myosin motor ensembles, whereas the prior model (Ref. 18) exclusively represents smooth trajectories. However, there is a lack of clarity regarding how stochasticity was introduced in the simulations, particularly given that both models employ the same mathematical framework. It would be beneficial if the authors could provide insights into the specific methodological variances between the two studies (the present one and Ref. 18).

We apologize for our lack of clarity and after rereading agree that this was a problem. In the revised manuscript we provide more details of the stochastic computational model used here and compare it to

the continuous model used in the Langmuir analysis

It appears that the smooth trajectory approach outlined in equation (18) effectively delineates the force generation phases. In contrast, the current methodology lacks a distinct transition or bifurcation between phases, particularly evident when plotting the (F-NMPD) relationship in the majority of simulated cases. Could you clarify the numerical criteria used to identify transition points between these force generation phases? Could you please offer additional information regarding the simulation protocol and the numerical algorithm employed to produce the results presented?

In our revised manuscript, we now carefully step through what is computationally occurring during each phase and how transitions between phases computationally occur.

We also describe our simulation protocol and numerical algorithms in much greater details.

Minor Points

Methods section: (Please review all the equations)

There is a typo (missing a minus sign) in equation (2) which should read as
$$F = -aN[\Delta G_0 + k_B T \ln(NAMD/NMDP)]/d$$

We thank reviewer 3 for catching this typo, and we have fixed it in the revised manuscript.

There is a typo. The temperature T is missing in the following expressions i.e, KB should be replaced by KBT. The two equations after equation 3.

We thank reviewer 3 for catching this typo, and we have fixed it in the revised manuscript.

Results section

“From simulations of a two-state binding reaction (Fig. 1), we plot time traces of F (Fig. 2B) and the occupancies of the AMD (NAMD, orange line) and MDP (NMDP, green line) states (Fig. 2C).” It appears that there is a misplacement in the citation order of Figures B and C. The traces for F are displayed in Figure C, whereas the occupancies are presented in Figure B.

We are grateful for reviewer 3 for catching this typo, and we have corrected it in the revised manuscript.

In Figure 2A, I suggest incorporating an arrow to denote the "Catastrophic" phase.

In the revised manuscript, we include this arrow.

In figure 3, what is the rationale behind choosing these particular values of the spring constants $k_{sys} = 0.01, 0.16, 2$ and 10 pN/nm. Please add a reference if possible.

In our revised manuscript, we now explain that the spring constant describes the effective stiffness of the ensemble system, which includes many factors (actin-myosin bound lifetimes, extracellular matrix, proteins that link actin and myosin filaments, etc) that make the effective stiffness highly variable. For example, the effects of varying the spring constant indicates how fibrosis qualitatively affects force generation. In our revised manuscript, we expand this discussion and include references.

In figures (3-7), it would be beneficial to display plots illustrating the state occupancies over time for all simulated cases, akin to the format shown in Figure 2B.

We made several attempts at this and were unable to get all of this data to fit neatly into one figure. However, we now include the state occupancies over time simulated in figures 3 – 7 in supplementary figures.

I highly suggest reconsidering the plotting of figures (3-7) by improving the scaling and opting for higher resolution. Adjusting the Y-axis limit would enhance the visibility of the data. Similar adjustments could be beneficial for the inset "a" figures as well.

We have improved the resolution of these figure and have adjusted scaling. When varying the y-axis limit between graphs there is a trade-off between improved visibility of data and diminished ability to compare absolute changes between graphs. In comparing fixed scale versus variable scale graphs, we believe that the fixed scale that allows a direct comparison between graphs is most informative.

Detecting the presence of the four loops in force generation can be challenging in certain scenarios. The transition or onset from one phase to another appears to be highly responsive to the range of parameters. It would be highly beneficial if you could share the simulation code, preferably in Matlab. Providing such code would greatly assist readers in gaining a clearer comprehension of the intricacies involved in the four loops of force generation.

In our latest submission, we now include the Python code that we used in our simulations.

July 30, 2024

Dr. Josh E Baker
University of Nevada, Reno School of Medicine
Pharmacology
1664 N. Virginia Street, MS 330
Reno, NV 89557

Re: 202313493R1

Dear Josh,

Thank you for submitting your manuscript, entitled "Stochastic Force Generation in an Isometric Binary Mechanical System" to JGP. Your manuscript has now been seen by 3 reviewers, whose comments are appended below. You will see that the reviewers consider the paper much improved, but also have raised several remaining concerns that should be addressed prior to further consideration of the manuscript at JGP. In particular, the comments with regards to how simulation results can be experimentally validated, the number of self citations versus citations to other peoples work, and the molecular explanation for how force changes along the isothermal axis with decreasing heads bound.

We would be pleased to receive a suitably revised manuscript that addresses these concerns. In addition, please do not hesitate to contact me (via the editorial office) if you feel that a discussion of the reviewers' and editors' comments would be helpful.

Please submit your revised manuscript via the link below along with a point-by-point letter that details your responses to the editors' and reviewers' comments, as well as a copy of the text with alterations highlighted (boldfaced or underlined). If the article is eventually accepted, it would include a 'revised date' as well as submitted and accepted dates. If we do not receive the revised manuscript within one year, we will regard the article as having been withdrawn. We would be willing to receive a revision of the manuscript at a later time, but the manuscript will then be treated as a new submission, with a new manuscript number.

Please pay particular attention to recent changes to our instructions to authors in the following sections: Data presentation, Blinding and randomization and Statistical analysis, under Materials and Methods, as shown here: <https://rupress.org/jgp/pages/submission-guidelines#prepare>. Re-review will be contingent on inclusion of the required information (including for data added during revision) and demonstration of the experimental reproducibility of the results. Also, to improve the reproducibility of published content, we have partnered with SciScore. Authors are prompted in eJP to copy and paste the Materials and Methods section of their manuscript for a SciScore assessment when submitting their revised manuscript. Authors are encouraged (not required) to further revise their Materials and Methods if the SciScore is below 4. More information can be found here: <https://rupress.org/jgp/pages/submission-guidelines#sciscore>

Please note, JGP now requires authors to submit Source Data used to generate figures containing gels and Western blots with all revised manuscripts (when applicable). This Source Data consists of fully uncropped and unprocessed images for each gel/blot displayed in the main and supplemental figures. If your paper includes cropped gel and/or blot images, please be sure to provide one Source Data file for each figure that contains gels and/or blots along with your revised manuscript files. File names for Source Data figures should be alphanumeric without any spaces or special characters (i.e., SourceDataF#, where F# refers to the associated main figure number or SourceDataFS# for those associated with Supplementary figures). The lanes of the gels/blots should be labeled as they are in the associated figure, the place where cropping was applied should be marked (with a box), and molecular weight/size standards should be labeled wherever possible.

Source Data files will be made available to reviewers during evaluation of revised manuscripts and, if your paper is eventually published in JGP, the files will be directly linked to specific figures in the published article.

Source Data Figures should be provided as individual PDF files (one file per figure). Authors should endeavor to retain a minimum resolution of 300 dpi or pixels per inch. Please review our instructions for export from Photoshop, Illustrator, and PowerPoint here: <https://rupress.org/jgp/pages/submission-guidelines#revised>

When revising your manuscript, please be sure it is a double-spaced MS Word file and that it includes editable tables, if appropriate.

Please submit your revised manuscript via this link:
Link Not Available

Thank you for the opportunity to consider your manuscript.

Sincerely,

Henk L. Granzier, Ph.D.
On behalf of Journal of General Physiology

Journal of General Physiology's mission is to publish mechanistic and quantitative molecular and cellular physiology of the highest quality; to provide a best-in-class author experience; and to nurture future generations of independent researchers.

Reviewer #1 (Comments to the Authors):

Overall this is a very much improved manuscript, clearer in many ways, which was my original problem.

I would require the authors to clearly show the values for F_0 and ΔG_0 in all figure legends used to generate the traces from Eq1 and Eq2.

I would invite the authors to offer a molecular explanation for how force increases along the isothermal axis with decreasing heads bound. Is it explained simply as the parallel spring per head attached relaxing as another head detaches?

Reviewer #2 (Comments to the Authors):

The rebuttal letter is unnecessarily strong, and long. In particular, unnecessarily long, because it spends much of the text (and of my time) defending the manuscript from a critic that I never did. I do not think that any of the scientists working in models of muscle mechanics (not me at least) wants to "reject thermodynamics" or to "disprove the second law of thermodynamics". The author is the only one that I know that asked "whether or not thermodynamics applies to muscle". I do not find in my review "objections to thermodynamics".

I do not agree with the author also about the possibility that "two models are mutually exclusive". Hypotheses can be mutually exclusive. Models are always approximations of the reality (i.e. wrong), and different choices about which aspects are more important to catch, can make the more model useful (or not). I just asked the authors to make this improvement more explicit if they wanted to keep the strong sentences used in the previous version. Finally, the authors is clearly allowed to test whatever they wants, but the predictions developed in the current manuscript are just a parametric analysis (as now recognized) as long as they are not tested experimentally, which in my opinion is not "how the model testing is accomplished".

To think that "many scientists [...] have rejected the possibility that thermodynamics applies to muscle" sounds a real conspiracy theory, but here I am probably doing the same error of the author and will not discuss more on it.

Going over the unnecessarily long parts of the rebuttal letter, the revised manuscript addresses many of the critical points that I have raised. Not last, it considerably toned down the unnecessarily strong statements about other models. In the new version of the manuscript, the macroscopic errors which previously precluded the possibility to understand the bases of the model itself are corrected. Also, in the current version, it is more clear the fact that an incremental research is presented, respect the "substantial body of research" developed in the previous 25 years by the author. Regard this point I have two minor critics. I personally find (minor point 1) that the amount of the manuscript spent in recalling the previous efforts of the author is too long, compared to the new results presented here. Similarly, I personally find (minor point 2) that there are different ways to avoid of using more than 40% of the bibliography for self-citation.

Major critics, which in my opinion should be solved before publication, are the following:

- 1) Significant statement should address more precisely the incremental research presented in the current manuscript
- 2) I truly believe that it is mandatory to cite examples of the so-called (only in this manuscript at the best of my knowledge) "corpuscular mechanic models". Despite toned down, in the current version the author does not address any specific model except the Huxley 1957 model (lines 118-120). In this way, the readers may have the wrong impression that any other model is "rejecting thermodynamics". Many models, which extends that seminal paper, included, in part or extensively, the thermodynamics in defining their theoretical bases, while others recognized their limits, and some don't. In lines 118-120, 335-336, 350-352, the authors should include references to some/several models present in the literature that they consider more representative of the "Corpuscular mechanics". In this way authors can defend their work and assumption. This could also open a wider debate on the topic.

Reviewer #3 (Comments to the Authors):

The authors have responded to and addressed all my comments. However, I would appreciate it if they could consider the following minor point:

Please include a brief paragraph on how the presented results can be validated using experimental data. Additionally, consider referencing other stochastic models published in this area (e.g., Aboelkassem et al., Biophysical Journal, 117(12), 2255-2272) and briefly discuss the usefulness of stochastic models in modeling muscle mechanics.

Responses to Reviewer Comments

Editor:

In particular, the comments with regards to **how simulation results can be experimentally validated, the number of self citations versus citations to other peoples work, and the molecular explanation for how force changes along the isothermal axis with decreasing heads bound.**

We, once again, greatly appreciate the time that all three reviewers have spent reviewing this manuscript. We understand that new theories can be difficult to work through. In the revised manuscript, we have addressed all of the remaining reviewer concerns and summarize these revisions here along with more detailed responses.

Molecular explanation for how force changes along the isothermal axis with decreasing heads bound.

According to a thermodynamic model, the Gibbs free energy for actin-myosin binding predicts that along the binding isotherm, muscle force varies with N_{AMD} and N_{MDP} as

$$F \propto -\Delta G^\circ + k_B T \cdot \ln[(N_{MDP}/(N_{AMD}+1))]. \quad \text{Eq. R1}$$

This is in contrast to corpuscular mechanics models that predict muscle force is determined from molecular forces (the definition of corpuscular mechanics) as

$$F = N_{AMD} \cdot F_{uni}, \quad \text{Eq. R2}$$

where F_{uni} is the average force of a bound myosin motor. With an increase in bound motors, Eq. 1 predicts that force decreases whereas Eq. 2 predicts that force increases.

Corpuscular mechanics assumes a molecular explanation for changes in muscle force (Eq. R2), whereas thermodynamics assumes an energetic explanation for changes in muscle force (Eq. R1). Thus, while it is reasonable to expect from a corpuscular mechanic perspective (Eq. 2) that we should be able to provide a molecular explanation for changes in force, thermodynamics (Eq. R1) does not provide one. The energetic term $k_B T \ln[(N_{MDP}+1)/N_{AMD}]$ in Eq. R1 is entropic (it describes the entropy of an ensemble of myosin motors) and so it has no molecular explanation (system entropy is not contained within individual molecules).

The figure on the right [Fig. 2C from (1)] illustrates an ensemble of 10 force-generating myosin switches. In the left panel, starting from an equilibrium ensemble of switches in state $N_{MDP} = 5$, $N_{AMD} = 5$, $F = 0$, a

force, $F_{uni} = \Delta F_{ext}$, is applied to the system (replace $k_{sys,d}$ with F_{uni}). The system responds (right panel) with the detachment of one switch, which decreases the force by F_{uni} , returning the system to its initial force ($F = 0$). However, the resulting ensemble in state $N_{MDP} = 6$, $N_{AMD} = 4$, $F = 0$ is now no longer at equilibrium because the number of microstates accessible in the left panel, $\Omega = 10!/(5!5!) = 252$, is greater than the number of microstates accessible in the right panel, $\Omega = 10!/(4!6!) = 210$, drawing the reaction back toward the larger number of microstates. The decrease in Ω (from left to right) corresponds to a decrease in entropy, $k_B \ln \Omega$, and an increase in free energy, $k_B T \ln \Omega$, from which we derive the entropic term in Eq. R1. In order for the ensemble of switches to equilibrate in state $N_{MDP} = 6$, $N_{AMD} = 4$, an initial force larger than F_{uni} must be applied to the system. This extra force is entropic (Eq. R1) and is required to prevent the ensemble from returning to the state, $N_{MDP} = 5$, $N_{AMD} = 5$, of maximum entropy. The molecular mechanism for this entropic force is dynamic, thermal, and not corpuscularian. All we know is that this extra entropic force is energetically required for the ensemble system to equilibrate in state $N_{MDP} = 6$, $N_{AMD} = 4$. As we apply more force, more motors detach, the number of microstates and entropy increases, and the entropic force required to balance the increased entropy increases. At a sufficiently large force, the ensemble equilibrates in state $N_{MDP} = 9$, $N_{AMD} = 1$, at which point the number of microstates is only $\Omega = 10$, entropy is at a minimum, and entropic free energy is at a maximum. At this point, the entropic force required to balance the entropic free energy is at a maximum even though only one myosin head is bound. This is described by Eq. R1 – the statistical mechanics of a binary system – and is the antithesis of corpuscular mechanics (Eq. R2).

The above describes only the entropic force (i.e., the change in force along the isotherm, Eq. R1). It does not describe the force that is balanced against the standard free energy, ΔG° , for actin-myosin binding (the offset of the isotherm that defines F_0 when $N_{MDP} = N_{AMD} = 5$).

How simulation results can be experimentally validated.

A thermodynamic muscle model is most directly validated by simultaneously measuring muscle force and the number of myosin motors in states AMD and MDP along the isotherm. Because the relationship between F and N_{AMD} in a thermodynamic model (Eq. R1) is completely opposite that in corpuscular mechanic models (Eq. R2), the observed correlation between F and N_{AMD} definitively distinguishes between these two models.

In 2000, we isolated the force-generating myosin step in skinned muscle by removing ATP and adding ADP and orthovanadate, an analog of P_i . We measured muscle force, F , with a force transducer while simultaneously measuring N_{AMD} spectroscopically while slowly increasing the force, F , applied to muscle. The figure on the right [Fig. 2 from (2)] is a plot of the observed fraction of N_{AMD} versus normalized force. These data clearly show that N_{AMD} decreases with increasing F and are accurately fit by Eq. R1 (solid line). In active isometric muscle, the observed correlation between F , N_{AMD} , and N_{MDP} at different P_i concentrations is similarly consistent with Eq. R1 (3). These experiments directly show that muscle is an entropic spring and provided the experimental basis for our thermodynamic model.

The dynamic force generated by small myosin ensembles like those measured by Kaya and colleagues provide a less direct validation. In these experiments, the amplitude, periodicity, stochasticity, and phases of force generation can be directly measured as a function of ATP, P_i , and ADP concentrations, optical trap stiffness, numbers of myosin motors etc. These measurements can then be compared with the predictions of simulations like those presented here. Periodic force generation emerges directly from an out-of-the-box two-state thermodynamic model, whereas, as described in our manuscript, periodic force generation is difficult to reconcile with corpuscular mechanic models.

Finally, existing experimental measurements of isometric force generation have all been interpreted using corpuscular mechanic models (Eq. 2). I look forward to analyzing these datasets with a thermodynamic model to determine if it provides simpler and more meaningful interpretations. There is indeed much work to be done in developing and validating a thermodynamic muscle model, and given that we are currently the only research group working on this problem progress is slow. After laying the formal foundation of a thermodynamic muscle model, my top priority is to compare simulated thermodynamic work loops with experimentally measured cardiac pressure-volume loops. Fortunately, an increasing number of researchers are realizing that muscle is a thermodynamic system and are becoming interested in further developing and validating this model, which should speed up the validation process.

The number of self-citations versus citations to other peoples' work.

The percent self-citations increased when we removed references to corpuscular mechanic models in our last revision in response to concerns that our challenges to corpuscular mechanic models were offensive and unnecessary. The remaining large percentage is not surprising considering that our research group has for 25 years been the only group developing and testing a thermodynamic muscle model (i.e., a model in which muscle force is defined by the free energy of a myosin ensemble, Eq. R1). Reviewer 2 now requires that we balance this out by adding back references to corpuscular mechanic models. We are pleased that Reviewer 2 sees this as an opportunity to better define corpuscular mechanics relative to thermodynamics and to open a wider debate on the subject, which we now do in the discussion section of our revised manuscript.

Reviewer #1 (Comments to the Authors):

Overall this is a very much improved manuscript, clearer in many ways, which was my original problem.

Thanks to the guidance of our reviewers!

I would require the authors to clearly show the values for F_0 and ΔG^0 in all figure legends used to generate the traces from Eq1 and Eq2.

Done. In the process I discovered and fixed a typo in the methods section. Units of ΔG^0 were written as $k_B T$ when they should have been RT .

I would invite the authors to offer a molecular explanation for how force increases along the isothermal axis with decreasing heads bound. Is it explained simply as the parallel spring per head attached relaxing as another head detaches?

Please see above. The increase in force has no molecular explanation. Consistent with a thermodynamic model, it has an energetic explanation. Specifically, it is an entropic force (Eq. R1). I summarize the description above in the discussion section of the revised manuscript.

Reviewer #2 (Comments to the Authors):

The rebuttal letter is unnecessarily strong, and long. In particular, unnecessarily long, because it spends much of the text (and of my time) defending the manuscript from a critic that I never did. I do not think that any of the scientists working in models of muscle mechanics (not me at least) wants to "reject thermodynamics" or to "disprove the second law of thermodynamics". The author is the only one that I know that asked "whether or not

thermodynamics applies to muscle". I do not find in my review "objections to thermodynamics".

We agree that no scientist would develop a model that intentionally violates physical laws, but this does not mean they don't inadvertently make assumptions that do. A thermodynamic muscle force is determined from a system free energy (see above, Eq. R1). A corpuscular mechanic muscle force is determined from molecular forces (Eq. R2). Rejecting a thermodynamic model because it is not a corpuscular mechanic model is rejecting thermodynamics. I apologize if we read too much into Reviewer 2's comments and are relieved that Reviewer 2 does not reject thermodynamics in defense of corpuscular mechanics.

I do not agree with the author also about the possibility that "two models are mutually exclusive". Hypotheses can be mutually exclusive. Models are always approximations of the reality (i.e. wrong), and different choices about which aspects are more important to catch, can make the more model useful (or not). I just asked the authors to make this improvement more explicit if they wanted to keep the strong sentences used in the previous version. Finally, the authors is clearly allowed to test whatever they wants, but the predictions developed in the current manuscript are just a parametric analysis (as now recognized) as long as they are not tested experimentally, which in my opinion is not "how the model testing is accomplished".

A model in which muscle force is determined from molecular forces (Eq. R2) and a model in which muscle force is determined from the free energy of a myosin ensemble (Eq. R1) are mutually exclusive models. It is absolutely not true that all models are equally wrong. This argument is used by corpuscularians to justify the free-for-all of fantastical and varied mechanisms that we see across corpuscular mechanic muscle models today. Models that violate physical laws are non-scientific and wrong. Models that are consistent with physical laws are imprecise.

To think that "many scientists [...] have rejected the possibility that thermodynamics applies to muscle" sounds a real conspiracy theory, but here I am probably doing the same error of the author and will not discuss more on it.

For 25 years scientists have rejected our manuscripts based solely on the argument that a thermodynamic model challenges their obsolete 17th century corpuscularian beliefs. This is the defining rationale of conspiracy theorists. I have never and would never reject a manuscript on the basis that the model being proposed challenges my beliefs.

Going over the unnecessarily long parts of the rebuttal letter, the revised manuscript addresses many of the critical points that I have raised. Not last, it considerably toned down the unnecessarily strong statements about other models. In the new version of the manuscript, the macroscopic errors which previously precluded the possibility to understand the bases of the model itself are corrected. Also, in the current version, it is

more clear the fact that an incremental research is presented, respect the "substantial body of research" developed in the previous 25 years by the author. Regard this point I have two minor critics. **I personally find (minor point 1) that the amount of the manuscript spent in recalling the previous efforts of the author is too long, compared to the new results presented here. Similarly, I personally find (minor point 2) that there are different ways to avoid of using more than 40% of the bibliography for self-citation.**

Removing this background would make the manuscript more difficult to understand for those who have not read our work and for those who are less familiar with thermodynamics. In the revised manuscript, we instead decrease the percentage of self-citations by increasing reference to the work of others.

Major critics, which in my opinion should be solved before publication, are the following:
1) Significant statement should address more precisely the incremental research presented in the current manuscript

In our revised manuscript, we improve our description of the context within which this current work is presented. Just as the hundreds of corpuscular mechanic models that have been published over the past 65 years are all incremental relative to other corpuscular mechanic models; the current manuscript is incremental relative to a thermodynamic model, expanding the testable formal foundation of this model. "Incremental" is relative, however. I recently had two different manuscripts rejected on the basis that a thermodynamic model is too radically different from conventional models with one reviewer's single closing argument being that the manuscript should be rejected because a thermodynamic model "would overturn decades of work". Not exactly "incremental" from this point of view.

2) I truly believe that it is mandatory to cite examples of the so-called (only in this manuscript at the best of my knowledge) "corpuscular mechanic models". Despite toned down, in the current version the author does not address any specific model except the Huxley 1957 model (lines 118-120). In this way, the readers may have the wrong impression that any other model is "rejecting thermodynamics". Many models, which extends that seminal paper, included, in part or extensively, the thermodynamics in defining their theoretical bases, while others recognized their limits, and some don't. In lines 118-120, 335-336, 350-352, the authors should include references to some/several models present in the literature that they consider more representative of the "Corpuscular mechanics". In this way authors can defend their work and assumption. This could also open a wider debate on the topic.

In the revised manuscript, we more clearly distinguish between thermodynamic and corpuscular mechanic models. We also address corpuscular mechanic models that incorporate muscle force, F , as a parameter, which I assume are the thermodynamic extensions of Huxley 57 to which Reviewer 2 refers.

Reviewer 2 can correct me if I'm wrong, but there are no models other than ours in which muscle force is defined by the free energy of an ensemble of myosin motors (Eq. R1). In other words, after A.V. Hill we are the only group that has published a thermodynamic muscle model. All other muscle models in the literature determine muscle force from molecular forces (Eq. R2). In other words, all other models are corpuscular mechanic models.

In 1938, A.V. Hill observed that muscle mechanics, energetics, and kinetics are functions of muscle force, F , and based on these observations he developed a thermodynamic equation that accurately describes the relationship between muscle force, F , muscle shortening velocities, V , muscle power output, and muscle heat output. In 1999, we observed that actin-myosin ATPase mechanics, kinetics, and energetics in muscle are functions of muscle force, F . Based on these observations we established that the molecular mechanism of muscle contraction is an entropic spring consisting of an ensemble of force-generating myosin switches. We then showed that this entropic spring bridges the gap between force-generating myosin switches and A.V. Hill's thermodynamic muscle force. It took the muscle field twenty more years before they began to realize that muscle force, F , affects muscle contraction. However, rather than arrive at the obvious conclusion that muscle is a thermodynamic system, corpuscularians have arbitrarily incorporated muscle force, F , into their corpuscular mechanic models as a parameter that influences a variety of new fantastical mechanisms. Arbitrarily inserting the parameter F into a model does not magically transform it into a thermodynamic model. Rather, it results in a nonsensical mixed scale model in which two mechanical state variables defined on completely different thermal (spatial and temporal) scales are both assumed to exist on one thermal scale.

Unconstrained by physical laws, it is unclear why corpuscularians stop at muscle and myosin forces and do not include in their models mechanical state variables defined on all thermal scales (the forces of tertiary, secondary, primary, and amino acid structures for every protein in muscle). If all models are equally wrong and the accuracy with which mathematical equations describe experimental data determines a model's usefulness, models that incorporate these additional mechanical parameters would surely be the most useful. If we care about physical constraints, however, a mechanical state variable can only be defined on the thermal scale at which that mechanical state is mechanically constrained.

According to thermodynamics, muscle force is mechanically constrained (defined) on only one thermal scale – the thermal scale of muscle. That is, muscle force, F , is the only mechanical state variable that can be defined in a model of muscle contraction. At this thermal scale the stochastic, thermally fluctuating forces of molecules on all smaller thermal scales (including those in myosin motors) are defined by muscle force, F , not the other way around. In other words, myosin motors function under the constraint of muscle force, F , with their kinetics and energetics defined by F not by forces that are somehow localized to and defined within individual myosin motors independent of muscle force.

We summarize these comparisons in our revised manuscript as required by Reviewer 2. We appreciate that Reviewer 2 requests this comparison not to challenge thermodynamics in defense of corpuscular mechanics but rather to inform the readership of the difference between these models in an effort to spark a wider debate on the topic. However, if Reviewer 2 takes offense to this comparison, we can once again remove it from our manuscript. One way or another, it has no impact on our analysis because scientifically the 17th century philosophy of corpuscular mechanics is obsolete and – disproven by Carnot 200 years ago – irrelevant to thermodynamics.

Reviewer #3 (Comments to the Authors):

The authors have responded to and addressed all my comments. However, I would appreciate it if they could consider the following minor point:

Please include a brief paragraph on how the presented results can be validated using experimental data.

Please see above, which is summarized in a revised paragraph in the discussion section.

Additionally, consider referencing other stochastic models published in this area (e.g., Aboelkassem et al., Biophysical Journal, 117(12), 2255-2272) and briefly discuss the usefulness of stochastic models in modeling muscle mechanics.

We reference and discuss such models and their usefulness in the revised discussion section of our manuscript.

September 4, 2024

Dr. Josh E Baker
University of Nevada, Reno School of Medicine
Pharmacology
1664 N. Virginia Street, MS 330
Reno, NV 89557

Re: 202313493R2

Dear Dr. Josh,

I am pleased to let you know that your manuscript, entitled "Stochastic Force Generation in an Isometric Binary Mechanical System" is scientifically acceptable for publication in Journal of General Physiology. Formal acceptance will follow when it is modified in accordance with the referees' remaining remarks (see below, the suggested text change we leave to your discretion) and our editorial policies.

Please note items that need attention are listed at the bottom of this email (under 'manuscript formatting checklist') and on the attached marked-up pdf file. Please also be sure to include a letter addressing the reviewers' comments point-by-point (if applicable) and a copy of the text with alterations highlighted (boldfaced or underlined). Your manuscript should be a double-spaced MS Word file and include editable tables, if appropriate.

Lastly, JGP requires a data availability statement for all research article submissions. These statements will be published in the article directly above the Acknowledgments. The statement should address all data underlying the research presented in the manuscript. Please visit the JGP instructions for authors for guidelines and examples of statements at <https://rupress.org/jgp/pages/editorial-policies#data-availability-statement>.

Please submit your final files via this link:
Link Not Available

Thank you for choosing to publish your research in JGP and please feel free to contact me with any questions.

Sincerely,

Henk L. Granzier, Ph.D.
On behalf of Journal of General Physiology

Journal of General Physiology's mission is to publish mechanistic and quantitative molecular and cellular physiology of the highest quality; to provide a best in class author experience; and to nurture future generations of independent researchers.

Manuscript formatting checklist:

- MS Word document of text needed (including editable tables)
- MS Word document of supplemental text needed, if applicable (including figure legends and editable tables)
- References need to follow JGP style (This article has numbered citations in it, which is not the style we use). Please refer to our guidelines here: <https://rupress.org/jgp/pages/reference-guidelines>)
- Brief Statement describing supplementary information needed, if applicable (in subsection at end of Materials & Methods)
- Please include a data availability statement preceding the Acknowledgments section. Please see <https://rupress.org/jgp/pages/editorial-policies#data-availability-statement>
- Figures created at sufficient resolution and in acceptable format (including supplemental if applicable). If working in Illustrator, we prefer .ai or .eps file format. If working in Photoshop please use 600dpi/1000dpi .tiff or .psd file format. Minimum resolution at estimated print size: Minimum resolution for all figures is 600 dpi. For figures that contain both photographs and line art or text, 600 dpi is highly recommended. Figures containing only black and white elements (line art, no color, and no gray) should be 1,000 dpi. Maximum figure size is 7 in wide x 9 in high (17.5 x 22.8 cm) at the correct resolution. <https://jgp.rupress.org/fig-vid-guidelines>
- Supplemental figures, if any, conforming to same guidelines as manuscript figures (noted above)
- If images resemble one from a prior publications, the author must seek permissions (to reproduce or adapt) from the original publisher. [You can resubmit your paper while waiting to hear back from the original publisher but please keep us updated]
- All authors must complete a disclosure form prior to acceptance. A link to complete the form has been sent to all coauthors.

Please provide the editorial office with updated email addresses if necessary

Reviewer #1 (Comments to the Authors):

My criticisms have been addressed.

Reviewer #2 (Comments to the Authors):

Like in the first rebuttal letter, the author seems much more interested in supporting the superiority of their models compared to those present in the literature than in highlighting the actual scientific results presented in this manuscript.

In the new version, the author substantially ignored my two minor points.

The authors addressed my first major criticism.

The author, intentionally or not, misinterpreted my request to add some references to the current models and used it to reintroduce a (useless and unsupported by facts) critique of a different approach which can (and in my opinion probably will) easily coexist with different modeling methods and hypotheses.

Therefore, I ask to erase lines 334-343.

Namely "Recently, it has become more widely accepted that muscle force, F , affects muscle
335 contraction; however, rather than arrive at the obvious conclusion that muscle is a
336 thermodynamic system, corpuscularians arbitrarily incorporate muscle force, F , into their
337 corpuscular mechanic models as a parameter that influences a variety of new fantastical
338 mechanisms (20, 34, 35). However, arbitrarily inserting the parameter F into a
339 corpuscular model does not magically transform it into a thermodynamic model. Rather,
340 it results in a nonsensical mixed scale model in which two mechanical state variables
341 defined on two completely different thermal scales are both assumed to exist on one
342 thermal scale. A mechanical state variable can only be defined on the thermal scale at
343 which that mechanical state is constrained."

Adding the two new references 34 and 35 in line 317 (with refs 27-32) should be enough. The reasons for this request are the same as discussed in my first revision.

Reviewer #3 (Comments to the Authors):

The authors have addressed all of my points and concerns.